# COMPLETE MULTI-MODAL METRIC LEARNING FOR MULTI-MODAL SARCASM DETECTION

## ABSTRACT

Multi-modal sarcasm detection identifies sarcasm from text-image pairs, an essential technology for accurately understanding the user's real attitude. Most research extracted the incongruity of text-image pairs as sarcasm information. However, these methods neglected inter-modal or intra-modal incongruities in fact and sentiment perspectives, leading to incomplete sarcasm information and biased performance. To address the above issues, this paper proposes a complete multi-modal metric learning network (CMML-Net) for multi-modal sarcasm detection tasks. Specifically, CMML-Net utilizes a fact-sentiment multi-task representation learning module to produce refined fact and sentiment text-image representation pairs. It then designs a complete multi-modal metric learning to iteratively calculate inter-modal and intra-modal incongruities in ~~fact and sentiment metric spaces~~ a unified space (e.g., fact and sentiment metric space), efficiently capturing complete multi-modal incongruities. CMML-Net performs well in explicitly capturing comprehensive sarcasm information and obtaining discriminative performance via deep metric learning. The state-of-the-art performance on the widely-used dataset demonstrates CMML-Net's effectiveness in multi-modal sarcasm detection.

## 1 INTRODUCTION

Sarcasm is a widely used implicit expression in which the real attitude conflicts with the literal meaning (Gibbs, 1986). This incongruity between real attitude and literal meaning is a crucial clue for identifying sarcastic intent (Joshi et al., 2015). Multi-modal sarcasm detection captures the user's real attitude by identifying incongruities across and within various modalities. It has a wide range of applications in social media monitoring and management, intelligent interactive systems, news dissemination, and public opinion analysis, etc. With the development of social media, it has garnered significant attention (Kolchinski & Potts, 2018; Desai et al., 2022).

Multi-modal sarcasm detection captures multi-modal incongruity from a fact perspective or sentiment perspective. Fact incongruity means sarcasm occurs when the literal meaning and the observed facts unexpectedly contrast (Grice, 1978; McDonald, 1999). Sentiment incongruity means sarcasm often occurs when the literal meaning is positive while the observed emotion is negative (Sperber & Wilson, 1987). Most research focuses on fact incongruity or sentiment incongruity. For example, Yue et al. (2023) leveraged prior knowledge from ConceptNet and contrastive learning to improve factual semantic incongruities in sarcasm detection tasks. Liang et al. (2022) constructed cross-modal graphs to extract sentiment incongruities and predict sarcasm. However, the lack of either the fact or sentiment perspective weakens sarcasm detection performance, as demonstrated in Figures 1 (a) and (b). Therefore, this paper focuses on mining complete incongruity features, including fact incongruity and sentiment incongruity.

Additionally, most multi-modal sarcasm detection research only focused on inter-modal incongruity as sarcasm information. For instance, Cai et al. (2019) and Xu et al. (2020) utilized cross-modal attention mechanisms and fusion strategies to discover factual semantic relevance in inter-modal contexts. Wang et al. (2024b) designed an align-fuse-collaborate mechanism to enhance the inter-modal incongruities in sarcasm information fusion. However, these works neglected the importance of intra-modal incongruity in sarcasm detection, leading to incomplete incongruities and biased performance. Moreover, intra-modal incongruity is more effective and provides supplementary information in some sarcastic scenarios, such as when the information between modalities is congruous

or when one of the modalities has too little information, as shown in the Figure 1 (c) and (d). Thus, this work focuses on an efficient capture strategy for complete multi-modal incongruities.

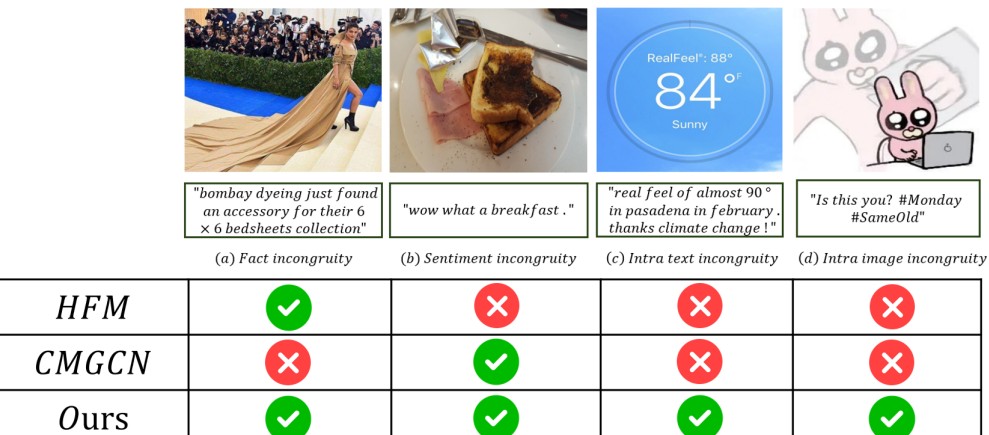

Figure 1: The test results of CMML-Net and different models on several examples including fact incongruity, sentiment incongruity, intra-modal text incongruity, and intra-modal image incongruity. HFM (Cai et al., 2019) utilized inter-modal attention to extra fact incongruity. CMGCN(Liang et al., 2022) built cross-modal graphs to extract sentiment incongruity.

This paper proposes a complete multi-modal metric learning network (CMML-Net) to efficiently capture complete multi-modal incongruity for sarcasm detection tasks. Specifically, CMML-Net utilizes a fact-sentiment multi-task representation learning module to extract fine-grained fact representations and sentiment representations from text-image data via Yolo-task and SenticNet-task. It then presents a fact-sentiment dual-stream network to construct fact and sentiment metric spaces for the following comprehensive incongruity capture. In the metric spaces, it designs complete multi-modal metric learning to iteratively calculate inter-modal and intra-modal incongruities in ~~fact and sentiment metric spaces~~ a unified space (e.g., fact and sentiment metric space). CMML-Net efficiently and explicitly captures complete multi-modal incongruities as effective sarcasm information and obtains comprehensive performance for multi-modal sarcasm detection. The state-of-the-art performance on the widely-used dataset demonstrates the superiority of CMML-Net in multi-modal sarcasm detection tasks. We released the codes and parameters to facilitate the research community. **[https://anonymous.4open.science/r/CMML-Net.]**

The main contributions of our paper can be summarized as follows:

- To our knowledge, the CMML-Net is the first work in multi-modal sarcasm detection to introduce deep metric learning to iteratively and explicitly calculate ~~complete multi-modal incongruities in fact and sentiment perspectives~~ inter-modal and intra-modal incongruities in a unified space (e.g., fact and sentiment metric space). It efficiently handles the biased performance of sarcasm detection models through comprehensive sarcasm information.
- CMML-Net is an effective fact-sentiment dual-stream framework in multi-modal sarcasm detection tasks. It contains a fact-sentiment multi-task representation learning module and fact-sentiment dual-stream network, gradually capturing complete multi-modal incongruities in fact and sentiment perspectives.

## 2 RELATED WORK

### 2.1 SINGLE-MODAL SARCASM DETECTION

Early research in sarcasm detection focused on text-based methods (Zhang et al., 2016; González-Ibánez et al., 2011; Riloff et al., 2013). SIARN (Tay et al., 2018) introduces an attention-based model to capture contrast and incongruity. SMSD (Xiong et al., 2019) utilizes a self-matching

network to explore word interactions. As research expanded, image-based sarcasm detection also gained attention. Studies on memes revealed that images alone could convey sarcasm, even without any textual content(Sharma et al., 2020; Maity et al., 2022). Certain visual features in the images play a critical role in expressing this sarcasm.

## 2.2 Multi-modal Sarcasm Detection

Multi-modal sarcasm detection has primarily focused on the image-text modality. Early works employed inter-modal attention to explore associations between modalities. For instance, HFM (Cai et al., 2019) utilized inter-modal attention to guide hierarchical modality fusion, while D&RNet (Xu et al., 2020) modeled semantic associations between images and text using inter-modal attention to capture commonalities and differences. Similarly, CMGCN (Liang et al., 2022) built cross-modal graphs to detect sarcasm by extracting sentiment incongruities. However, these works primarily emphasize inter-modal incongruities, leaving intra-modal incongruities underexplored.

Some studies have recognized the importance of intra-modal incongruities. For instance, Att-Bert (Pan et al., 2020) and Multi-View CLIP (Qin et al., 2023) employed separate attention mechanisms, and InCrossMGs (Liang et al., 2021) used distinct GCNs to extract inter-modal and intra-modal incongruities separately. However, they focus only on factual semantic incongruities. Moreover, DMSD-CL Jia et al. (2024) and G2SAM(Wei et al., 2024) learn sarcasm patterns through sample-level differences.

In addition to these models, more nuanced incongruity detection frameworks have emerged. DIP (Wen et al., 2023) proposed a dual-stream architecture with semantic and sentiment streams to identify inter-modal incongruities. FSICN (Lu et al., 2024) introduced a fact-sentiment dual-stream network to capture incongruities. Another recent work, KnowleNet (Yue et al., 2023), improved factual incongruity detection using prior knowledge from ConceptNet and contrastive learning. In contrast, MuMu (Wang et al., 2024b) designed an align-fuse-collaborate mechanism to enhance inter-modal incongruity detection. Despite these advancements, these models remain focused on inter-modal incongruities.

CMML-Net, our proposed model, addresses these gaps by explicitly and efficiently capturing complete multi-modal incongruities in fact and sentiment perspectives via deep metric learning.

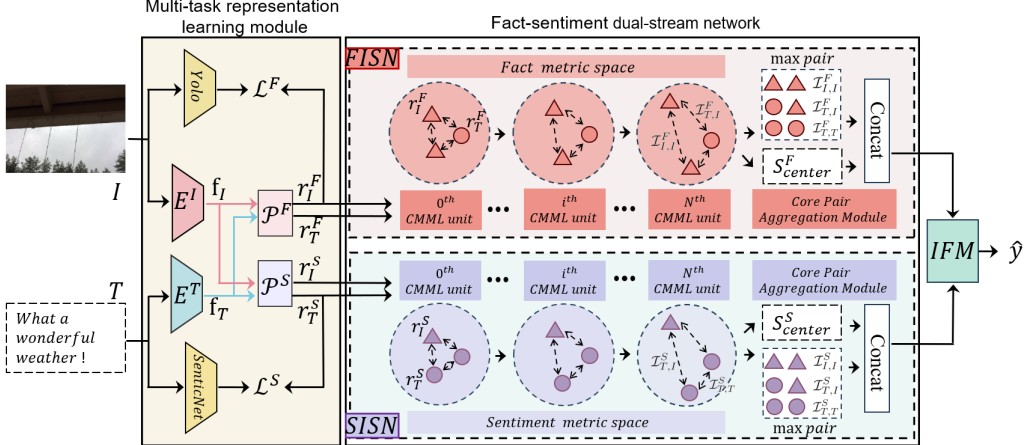

Figure 2: The overall architecture of CMML-Net.

## 3 Proposed method

There are many sarcastic situations in multi-modal sarcasm detection tasks, which can be easily missed and lead to biased recognition performance. Therefore, we propose a complete multi-modal metric learning network to efficiently capture complete multi-modal incongruity for multi-modal sarcasm detection tasks. The framework of CMML-Net is shown in Figure 2. CMML-Net comprises

a fact-sentiment multi-task representation learning module, a fact-sentiment dual-stream network, and an incongruity fusion module (IFM). The details of each module are as follows.

Multi-modal sarcasm detection aims to determine whether a given text-image pair conveys sarcasm. for a dataset $D$ consisting of multi-modal samples, each sample $d \in D$ includes a sentence $T$ with $n$ words $\{t_1, t_2, t_3, \ldots, t_n\}$ and an image $I$ divided into $m$ patches $\{v_1, v_2, v_3, \ldots, v_m\}$.

### 3.1 FACT-SENTIMENT MULTI-TASK REPRESENTATION LEARNING MODULE

The fact-sentiment multi-task representation learning module aims to produce refined fact and sentiment text-image representation pairs for the following complete multi-modal incongruity capture. It leverages multi-task learning to construct fact and sentiment representation spaces. It mainly consists of fact text-image representation extraction and sentiment text-image representation extraction.

~~We utilize the pre-trained and frozen CLIP text and image encoders to obtain public task-agnostic general features (Radford et al., 2021). Specifically,~~ The text encoder maps $T$ into word-level embeddings $\mathbf{f}_T = \text{CLIP}_{\text{text}}(T)$, and the image encoder transforms $I$ into patch-level embeddings $\mathbf{f}_I = \text{CLIP}_{\text{image}}(I)$.

Fact text-image representation extraction obtains effective fact text-image representations via Yolo-task. Yolo queries object existence vectors $\mathbf{y}^F \in [0, 1]^k$ from the input image $I$, where $k$ is the number of detected object categories and the superscript capital letter $F$ stands for fact. The vectors guide the fact projection layer $\mathcal{P}^F$ to project the image features $\mathbf{f}_I$ into fact image representation $\mathbf{r}_I^F$. The shared fact projection layer similarly projects the text features $\mathbf{f}_T$ into fact text representation $\mathbf{r}_T^F$. Both representations are implicitly aligned in a unified fact representation space $\mathcal{S}^F$, which facilitates fine-grained factual semantic inference. The binary cross-entropy loss (BCE) optimizes the projection by minimizing the difference between the predicted object existence vector $\hat{\mathbf{y}}^F$ and the pseudo labels $\mathbf{y}^F$:

$$\mathcal{L}^F = -\sum_{i=1}^{k} \left( \mathbf{y}_i^F \log \hat{\mathbf{y}}_i^F + (1 - \mathbf{y}_i^F) \log(1 - \hat{\mathbf{y}}_i^F) \right) \tag{1}$$

Sentiment text-image representation extraction obtains effective sentiment text-image representations via SenticNet-task. SenticNet queries the input text and generates a sentiment polarity vector $\mathbf{y}^S \in [-1, 1]^n$. Here, $n$ denotes the number of tokens, $y_i^S = 0$ indicates no sentiment match (Liang et al., 2022), and the superscript capital letter $S$ stands for sentiment. The vectors guide the sentiment projection layer $\mathcal{P}^S$ to project the text features $\mathbf{f}_T$ into sentiment text representation $\mathbf{r}_T^S$. The shared sentiment projection layer similarly projects the image features $\mathbf{f}_I$ into sentiment image representation $\mathbf{r}_I^S$. Both representations are implicitly aligned in the unified sentiment representation space $\mathcal{S}^S$. The mean squared error (MSE) loss minimizes the prediction error.

$$\mathcal{L}^S = \frac{1}{n} \sum_{i=1}^{n} (y_i^S - \hat{y}_i^S)^2 \tag{2}$$

### 3.2 FACT-SENTIMENT DUAL-STREAM NETWORK

The fact-sentiment dual-stream (FSDS) network aims to capture intra-modal and inter-modal incongruities of fact text-image representations and sentiment text-image representations in unified metric spaces. This obtains complete multi-modal incongruity and comprehensive recognition performance for sarcasm detection tasks. FSDS network presents fact incongruity sub-network (FISN) and sentiment incongruity subnetwork (SISN) to construct fact and sentiment metric spaces. It then designs $N$ complete multi-modal metric learning units to iteratively and explicitly calculate complete multi-modal incongruities in ~~fact and sentiment metric spaces~~ a unified space (e.g., fact and sentiment metric space). FSDS network efficiently captures comprehensive fact and sentiment sarcasm information for multi-modal sarcasm detection tasks via deep metric learning.

#### 3.2.1 FACT INCONGRUITY SUB-NETWORK

Facts describe the existence of objects or events, which are hidden in semantics. Fact incongruity refers to the incongruity between factual semantic information in multi-modal data. FISN aims to

capture intra-modal and inter-modal incongruities in fact metric spaces as complete multi-modal fact incongruities. Inspired by deep metric learning, we designed the complete multi-modal metric learning (CMML). CMML iteratively calculates inter-modal and intra-modal incongruities in the fact metric space via $N$ CMML units, which is the designed basic component of CMML.

Each CMML unit computes the incongruity of all text-text, text-image, and image-image representation pairs in a unified fact metric space, and updates them via dynamic separation and non-linear adjustment to obtain more discriminative incongruity representations. Thus, it will gradually obtain complete multi-modal incongruities after $N$ CMML units. It mainly consists of initial incongruity computation, dynamic separation, and non-linear adjustment.

**Initial incongruity computation** aims to align the unified fact representation space $\mathcal{S}^F$ into the unified fact incongruity metric space $(\mathcal{S}^F, \mathcal{I}^F)$. It computes initial incongruity by measuring the Euclidean distance between each pair of representations. The representation pairs in the unified space $\mathcal{S}^F$ include text-text, text-image, and image-image. Larger distances indicate higher degrees of incongruity. This establishes a clear reference for reliable updating of the representation of the complete multi-modal incongruity. The incongruity between $\mathbf{r}_u^F \in \mathcal{S}^F$ and $\mathbf{r}_v^F \in \mathcal{S}^F$ can be calculated as follows:

$$\mathcal{I}_{u,v}^F = \|\mathbf{r}_u^F - \mathbf{r}_v^F\| \tag{3}$$

**Dynamic separation** enhances the discriminative performance by increasing the separation between incongruous representations. Each representation $\mathbf{r}_u^F \in \mathcal{S}^F$ receives directional clues from other representation. We employ adaptive weights to reinforce representation itself by relatively congruous representations and prevent assimilation from its incongruous representations. The adaptive weights between each pair of representation $\mathbf{r}_u^F \in \mathcal{S}^F$ and $\mathbf{r}_v^F \in \mathcal{S}^F$ in the $i$-1-th CMML unit are calculated as follows:

$$\mathbf{w}_{u,v}^{F,(i-1)} = \frac{\exp\left(-(\mathcal{I}_{u,v}^{F,(i-1)})^2\right)}{\sum_{\mathbf{r}_{v'}^{F,(i-1)} \in \mathcal{S}^{F,(i-1)}} \exp\left(-(\mathcal{I}_{u,v'}^{F,(i-1)})^2\right)} \tag{4}$$

followed by the representation $\mathbf{r}_u^F \in \mathcal{S}^F$ in the $i$-1-th CMML unit update:

$$\mathbf{r}_u^{F,(i-1)'} = \mathbf{r}_u^{F,(i-1)} + \sum_{\mathbf{r}_v^F \in \mathcal{S}^{F,(i-1)}} \mathbf{w}_{u,v}^{F,(i-1)} (\mathbf{r}_v^{F,(i-1)} - \mathbf{0}) \tag{5}$$

**Non-linear adjustment** enhances the discriminative performance by refining the topology of the fact metric space. It leverages a simple yet effective deep learning approach to handle complex incongruity relationships and improve the robustness of incongruity discrimination. The representation $\mathbf{r}_u^F \in \mathcal{S}^F$ in the $i$-1-th CMML unit update:

$$\mathbf{r}_u^{F,(i)} = FFN\left(\mathbf{r}_u^{F,(i-1)'}\right) \tag{6}$$

After the iterative calculating of $N$ CMML units, the distance between strongly incongruous representations increases more significantly than that between weakly incongruous ones. FISN obtains more discriminative incongruity representations.

The core pair aggregation module (CPAM) aggregates the fact sarcasm information to output. It firstly screens the target incongruity in the fact incongruity metric space to capture complete multi-modal fact incongruity. The incongruous representation pairs of the fact metric space are summarized into three categories: intra-modal image pairs $(\mathbf{r}_I^F, \mathbf{r}_I^F)$, inter-modal pairs $(\mathbf{r}_T^F, \mathbf{r}_I^F)$, and intra-modal text pairs $(\mathbf{r}_T^F, \mathbf{r}_T^F)$. After selecting the most incongruous pair in each category, we further select the overall most incongruous pair. It helps the CMML-Net focus on the most significant signal that is most likely to convey sarcasm. The unified metric space captures incongruities in just one efficient step. The capturing in the CPAM is formalized as follows:

$$(\mathbf{r}_{u'}^F, \mathbf{r}_{v'}^F) = \underset{(\mathbf{r}_u^F, \mathbf{r}_v^F) \in \mathcal{S}^F \times \mathcal{S}^F}{\arg\max} (\mathcal{I}_{I,I}^F, \mathcal{I}_{T,I}^F, \mathcal{I}_{T,T}^F) \tag{7}$$

CPAM takes the overall most incongruous pair as local information and fuses it with the center of the global representation space to produce an embedding of FISN called fact incongruity (FI).

$$\mathbf{f}_{FI} = (\mathbf{r}_{u'}^F, \mathbf{r}_{v'}^F) \oplus \mathcal{S}_c^F \tag{8}$$

### 3.2.2 SENTIMENT INCONGRUITY SUB-NETWORK

Sarcasm often implicitly expresses dissatisfaction in a positive manner. For instance, the statement "What a wonderful day!" alongside an image of a rainy day produces a sentiment incongruity. The word "wonderful" expresses a positive sentiment, but the rain in the image shows a negative impression. SISN aims to capture intra-modal and inter-modal incongruities in sentiment metric spaces as complete multi-modal sentiment incongruities.

Similar to FISN, SISN uses CMML to capture complete multi-modal incongruities via $N$ CMML units. Since CMML units in both FISN and SISN perform metric learning in metric spaces, we leverage the perspectives of the metric space to distinguish the type of incongruities being captured.

Each CMML unit operates within the sentiment metric space $(\mathcal{S}^S, \mathcal{I}^S)$. It mainly consists of initial incongruity computation, dynamic separation, and non-linear adjustment.

In **initial incongruity computation**. The incongruity between $\mathbf{r}_u^S \in \mathcal{S}^S$ and $\mathbf{r}_v^S \in \mathcal{S}^S$ can be calculated as follows:

$$\mathcal{I}_{u,v}^S = \|\mathbf{r}_u^S - \mathbf{r}_v^S\| \tag{9}$$

In **dynamic sepration**, The adaptive weights between each pair of representation $\mathbf{r}_u^S \in \mathcal{S}^S$ and $\mathbf{r}_v^S \in \mathcal{S}^S$ in the $i$-1-th CMML unit are calculated as follows:

$$\mathbf{w}_{u,v}^{S,(i-1)} = \frac{\exp\left(-(\mathcal{I}_{u,v}^{S,(i-1)})^2\right)}{\sum_{\mathbf{r}_{v'}^{S,(i-1)} \in \mathcal{S}^{S,(i-1)}} \exp\left(-(\mathcal{I}_{u,v'}^{S,(i-1)})^2\right)} \tag{10}$$

followed by the representation $\mathbf{r}_u^S \in \mathcal{S}^S$ in the $i$-1-th CMML unit update:

$$\mathbf{r}_u^{S,(i-1)'} = \mathbf{r}_u^{S,(i-1)} + \sum_{\mathbf{r}_v^S \in \mathcal{S}^{S,(i-1)}} \mathbf{w}_{u,v}^{S,(i-1)}(\mathbf{r}_v^{S,(i-1)} - \mathbf{0}) \tag{11}$$

In **non-linear adjustment**, the representation $\mathbf{r}_u^S \in \mathcal{S}^S$ in the $i$-1-th CMML unit update:

$$\mathbf{r}_u^{S,(i)} = FFN\left(\mathbf{r}_u^{S,(i-1)'}\right) \tag{12}$$

After the iterative calculating of $N$ CMML units, SISN obtains more discriminative incongruity representations.

The core pair aggregation module (CPAM) aggregates the sentiment sarcasm information to output. The incongruous representation pairs of the sentiment metric space are summarized into three categories: intra-modal image pairs $(\mathbf{r}_I^S, \mathbf{r}_I^S)$, inter-modal pairs $(\mathbf{r}_T^S, \mathbf{r}_I^S)$, and intra-modal text pairs $(\mathbf{r}_T^S, \mathbf{r}_T^S)$. The capturing in the CPAM is formalized as follows:

$$(\mathbf{r}_{u'}^S, \mathbf{r}_{v'}^S) = \arg\max_{(\mathbf{r}_u^S, \mathbf{r}_v^S) \in \mathcal{S}^S \times \mathcal{S}^S} (\mathcal{I}_{I,I}^S, \mathcal{I}_{T,I}^S, \mathcal{I}_{T,T}^S) \tag{13}$$

The sentiment incongruity (SI) of SISN is computed as:

$$\mathbf{f}_{SI} = (\mathbf{s}_{u'}^S, \mathbf{s}_{v'}^S) \oplus \mathcal{S}_c^S \tag{14}$$

### 3.3 INCONGRITY FUSION MODULE

We fuse the FI denoted as $\mathbf{f}_{FI}$ with the SI denoted as $\mathbf{f}_{SI}$ as the final incongruity and send it to the prediction layer. The total loss function balances the contributions of the fact-related loss, sentiment-related loss, and prediction loss:

$$\mathcal{L} = \alpha \mathcal{L}^F + \alpha \mathcal{L}^S + (1 - 2\alpha)\mathcal{L}^{pred} \tag{15}$$

## 4 EXPERIMENT

### 4.1 DATASET AND EVALUATION METRICS

We conduct experiments on the publicly available **M**ultimodal **S**arcasm **D**etection (MSD) dataset (Cai et al., 2019). Each sample in the dataset consists of text-image pairs. Samples expressing sarcasm are labeled as positive, and those without sarcasm are labeled as negative. Following previous works (Cai et al., 2019; Xu et al., 2020; Liang et al., 2022), we report accuracy, precision, recall, binary-average, and macro-average results for evaluation.

## 4.2 IMPLEMENTATION DETAILS

We used the CLIP ViT-B/32 model (Radford et al., 2021) with frozen parameters for unified token-level image and text feature extraction. For the fact stream, we employed YOLO v10-s with frozen parameters (Wang et al., 2024a). The CMML-Net was trained using AdamW with a learning rate set to 1e-4, weight decay at 1e-4, and $\alpha$ at 7.5%, over ten epochs.

## 4.3 BASELINE MODELS

To evaluate the performance of CMML-Net, we compare it against several state-of-the-art baselines, categorized into image-modality, text-modality, and multi-modal methods. **Image-modality methods**: ResNet (Cai et al., 2019), ViT (Dosovitskiy, 2020). **Text-modality methods**: Bi-LSTM (Graves & Schmidhuber, 2005), SIARN (Tay et al., 2018), SMSD (Xiong et al., 2019), BERT (Kenton & Toutanova, 2019). **Multi-modal methods**: HFM (Cai et al., 2019), InCrossMGs (Liang et al., 2021), CMGCN (Liang et al., 2022), Att-BERT (Pan et al., 2020), DIP (Wen et al., 2023), KnowleNet (Yue et al., 2023), FSICN (Lu et al., 2024), Mumu (Wang et al., 2024b) , Multi-view CLIP (Qin et al., 2023), DMSD-CL (Jia et al., 2024), G2SAM (Wei et al., 2024). The details of these methods have been described in the related work section.

## 4.4 MAIN RESULT

Table 1 presents the performance comparison of our proposed method against other state-of-the-art (SOTA) approaches. CMML-Net achieves the best performance across all evaluation metrics, demonstrating its ability to more comprehensively and accurately capture sarcasm information. Specifically, CMML-Net outperforms unimodal approaches due to the complementary nature of multi-modal data. Among unimodal methods, text-based approaches perform better than image-based ones, largely because of the abstract nature and implicit details in images. Therefore, CMML-Net employs a Yolo-task representation extraction to learn fine-grained factual semantics from images.

CMML-Net outperforms existing SOTA method MuMu (Wang et al., 2024b) by a significant margin—achieving +1.4% (90.73% vs. 92.04%), +1.8% (88.62% vs. 90.25%), and +1.5% (90.40% vs. 91.76%) on accuracy, binary-F1, and macro-F1 scores, respectively. Although MuMu utilizes the same feature extractor as ours, MuMu is limited because of the incomplete multi-modal incongruities. The performance gap between CMML-Net and these models highlights the effectiveness of CMML-Net in capturing complete multi-modal incongruities in fact and sentiment perspectives.

## 4.5 ABLATION STUDY

We conducted an ablation study to assess the impact of each module, as shown in Table 2. First, the absence of FISN and SISN also led to performance degradation. It demonstrates the biased performance of capturing multi-modal incongruity only from a fact perspective or sentiment perspective.

Additionally, Removing the Yolo-task fact representation extraction led to a drop in accuracy and macro-F1, underscoring the importance of learning fine-grained factual semantics via Yolo-task. Yolo-task improves the quality of fact represen-

Table 2: Ablation study results.

| Model Name | ACC(%) | F1(%) |
|---|---|---|
| w/o FISN | 91.23 | 90.87 |
| w/o SISN | 91.28 | 90.89 |
| w/o Yolo-task | 91.19 | 90.85 |
| w/o CMML units | 90.98 | 90.68 |
| Full Model | **92.04** | **91.76** |

tation space and prevents the performance from decreasing because of the collapse between fact representation and sentiment representation. In high-quality and separate fact and sentiment metric space, CMML-Net achieves more significant performance gains by deep metric learning.

Ablating CMML units showed further performance reductions. CMML units play a critical role in calculating complete multi-modal incongruity in fact and sentiment metric space. The model with CMML units obtains powerful discriminative performance and handles the biased performance by capturing complete multi-modal incongruity in fact and sentiment perspectives.

| Modality | Model | ACC(%) | Binary-Average | | | Macro-Average | | |
|---|---|---|---|---|---|---|---|---|
| | | | P(%) | R(%) | F1(%) | P(%) | R(%) | F1(%) |
| Image | Resnet | 64.76 | 54.41 | 70.80 | 61.53 | 60.12 | 73.08 | 65.97 |
| | ViT | 67.83 | 57.93 | 70.07 | 63.43 | 65.69 | 71.35 | 68.40 |
| Text | Bi-LSTM | 81.90 | 76.66 | 78.42 | 77.53 | 80.97 | 80.13 | 80.55 |
| | SIARN | 80.57 | 75.55 | 75.70 | 75.63 | 80.34 | 78.81 | 79.57 |
| | SMSD | 80.90 | 76.46 | 75.18 | 75.82 | 80.87 | 78.20 | 79.51 |
| | BERT | 83.85 | 78.72 | 82.27 | 80.22 | 81.31 | 80.87 | 81.09 |
| Image+Text | HFM | 86.63 | 83.84 | 84.18 | 84.01 | 86.24 | 86.28 | 86.26 |
| | InCrossMGs | 86.10 | 81.38 | 84.36 | 82.84 | 85.39 | 85.80 | 85.60 |
| | CMGCN | 87.55 | 83.63 | 84.69 | 84.16 | 87.02 | 86.97 | 87.00 |
| | Att-Bert | 86.05 | 78.63 | 83.31 | 80.90 | 80.87 | 85.08 | 82.92 |
| | DIP | 89.59 | 87.76 | 86.58 | 87.17 | 88.46 | 89.13 | 89.01 |
| | KnowleNet | 88.87 | 88.59 | 84.18 | 86.33 | 88.83 | 88.21 | 88.51 |
| | Multi-view CLIP | 88.33 | 82.66 | 88.65 | 85.55 | - | - | - |
| | DMSD-CL | 88.95 | 84.89 | 87.90 | 86.37 | 88.35 | 88.77 | 88.54 |
| | G2SAM | 90.48 | 87.95 | 89.02 | 88.48 | 89.44 | 89.79 | 89.65 |
| | FSICN | 90.55 | _89.93_ | _89.51_ | _89.72_ | 90.16 | _90.42_ | 90.29 |
| | MuMu | _90.73_ | 88.81 | 88.44 | 88.62 | _90.43_ | 90.37 | _90.40_ |
| | **Ours⋆** | **92.04⋆** | **90.21⋆** | **90.30⋆** | **90.25⋆** | **91.75⋆** | **91.77⋆** | **91.76⋆** |

Table 1: Results with ⋆ denote the significance tests of our CMML-Net over the baseline models at p-value < 0.01. The best results are highlighted in boldface, while the second-best results are underlined.

### 4.6 IMPACT OF YOLO-TASK REPRESENTATION EXTRACTION ON REPRESENTATION SPACE

We conducted an experiment to evaluate how Yolo-task representation extraction influences the separation of initial fact and sentiment representation spaces. To our knowledge, none of the existing fact-sentiment dual-stream networks research explicitly guides representations of fact sub-network in multi-modal sarcasm detection (Wen et al., 2023; Lu et al., 2024). We applied UMAP for dimensional reduction while preserving topology (Figure 3). The left figure shows the fact-sentiment representation space without the guidance information produced from the Yolo-task, and the right figure shows the fact-sentiment representation space after introducing Yolo-task representation extraction.

By analyzing the metric spaces, we observe that the lack of guidance information produced from the Yolo-task causes the FISN to collapse into the SISN. This limits the FISN's ability to capture incongruity at the fine-grained factual semantics level. This also weakens the SISN's contribution and leads to overall performance degradation. The Yolo-task representation learning module prevents this collapse and enhances the model's ability to maintain distinct representation spaces.

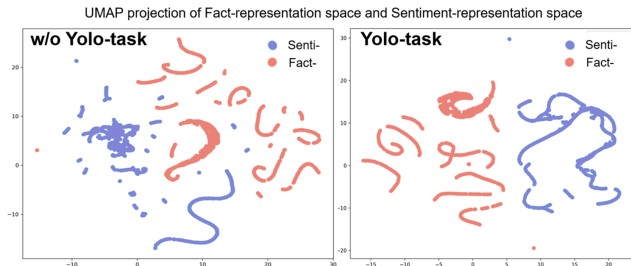

Figure 3: The fact-sentiment representation space. Model without Yolo-task (left) vs. with Yolo-task (right).

In summary, Yolo-task representation extraction significantly improves the separation between fact and sentiment representation spaces. We further validated this by training GAN models to distinguish between the two spaces. The model with Yolo-task representation extraction achieved a much higher accuracy of 93.17%, compared to 66.52% without Yolo-task representation extraction, confirming a 26.59% improvement in performance. This further demonstrates the effectiveness of Yolo-task representation extraction in preserving the integrity of FISN and SISN.

## 4.7 ROLE OF CMML UNITS IN CAPTURING COMPLETE MULTI-MODAL INCONGRUITY

We designed an experiment to verify the role of the CMML units in capturing complete multi-modal incongruities by tracking the performance of the full model and the model without the CMML units during training (shown in Figure 4). The model without CMML units lacks the mechanism to iteratively calculate inter-modal and intra-modal incongruities in the fact and sentiment metric space. It captures incongruities with the initial discriminative performance. The stacked bar chart displays the proportions of sentiment and fact incongruities, which are further divided into inter-modal, intra-image, and intra-text incongruities.

The data shows that both models exhibit similar performance trends in the early training stages. It indicates that initial optimization depends on

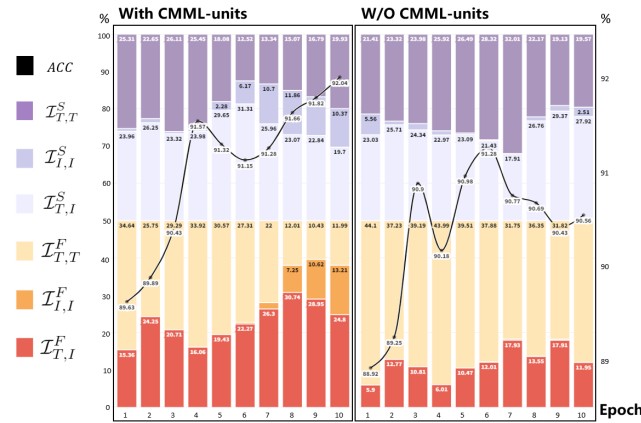

Figure 4: The proportion of max incongruity representation from inter-modal or intra-modal perspectives in both fact and sentiment perspectives over different epochs. Model with CMML units (Left) vs. without CMML units (Right).

global information from metric space. However, as training progresses and the models reach the first performance peak, the full model enters a refinement phase driven by CMML units. During this phase, $N$ CMML units iteratively calculate inter-modal and intra-modal incongruities in the fact and sentiment metric space. The model obtains more discriminative incongruity representations and efficiently captures complete multi-modal incongruity. The proportion of intra-image incongruities rises and eventually matches that of intra-text incongruities. In contrast, the model without CMML units stagnates, with its performance fluctuating slightly. This demonstrates that CMML units are crucial for handling biased performance through comprehensive sarcasm information, especially in later training stages.

## 4.8 MODEL EFFICIENCY ANALYSIS

We conducted experiments to validate the efficiency of CMML in capturing complete multi-modal incongruities in a unified metric space. We also compare it with the prior methods that have potential ability to capture complete multi-modal incongruities through reconstructing the framework. DIP (Wen et al., 2023) identifies inter-modal incongruities using Gaussian distribution differences, but its mechanism cannot be directly applied to intra-modal incongruities. Due to some practical limitations, it is difficult for us to reproduce FSICN (Lu et al., 2024). Att-Bert (Pan et al., 2020) employed separate attention mechanisms, and InCrossMGs (Liang et al., 2021) used distinct GCNs to extract inter-modal and intra-modal incongruities separately. This leads to significant computational overhead.

Table 3: Model Efficiency Comparision.

| Model | Performance | | Backend Model | |
|---|---|---|---|---|
| | ACC(%) | F1(%) | Params | GFLOPS |
| Att-Dual | 90.98 | 90.65 | 105.7M (↑511%) | 204.4 (↑527%) |
| GCN-Dual | 90.39 | 90.06 | 49.4M (↑186%) | 127.3 (↑290%) |
| Ours | **92.04** | **91.76** | **17.3M** | **32.6** |

For a fair comparison, we replicated previous models based on GCN and attention mechanisms (Pan et al., 2020; Liang et al., 2021) within our dual-stream framework and excluded the common frontend architecture (CLIP and Yolo) to focus on the backend.

As shown in Table 3, CMML-Net demonstrates a significant reduction in computational complexity, using 65% fewer parameters and 74% fewer FLOPS compared to the GCN-dual-stream model and 83% fewer parameters and 84% fewer FLOPS compared to the Attention-dual-stream model. These results confirm CMML's ability to maintain high performance with much lower resource demands.

## 5 CONCLUSION

In this paper, we propose CMML-Net, the first work in multi-modal sarcasm detection to introduce deep metric learning to explicitly and efficiently capture complete multi-modal incongruities in fact and sentiment perspectives. In the fact-sentiment multi-task representation learning module, we use Yolo-task and SenticNet-task representation extraction to produce refined fact and sentiment text-image representation pairs, respectively. It enhances the model's ability to maintain distinct representation spaces. In the fact-sentiment dual-stream network consisting of FISN and SISN, we designed a CMML to iteratively calculate inter-modal and intra-modal incongruities in fact and sentiment perspectives a unified space (e.g., fact and sentiment metric space) by a complete multi-modal metric learning. It then explicitly and efficiently captures complete multi-modal incongruities. CMML-Net handles the biased performance of sarcasm detection models by comprehensive sarcasm information. The state-of-the-art performance on the widely-used dataset demonstrates CMML-Net's superiority.

## 6 REPRODUCIBILITY STATEMENT

We provide a detailed description of the implementation in Section 4.2. Additionally, we have open-sourced our codebase to facilitate the reproduction of our results, which are available on the project website: **[https://anonymous.4open.science/r/CMML-Net-873E]**.

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

## A EXAMPLES OF THE INCONGRUITY IN SARCASTIC REMARKS

We demonstrate the effectiveness of capturing complete multi-modal incongruity in fact and sentiment perspectives through some specific sarcastic remarks (Figure 5). Incomplete incongruity will lead to biased performance due to the lack of sarcastic information.

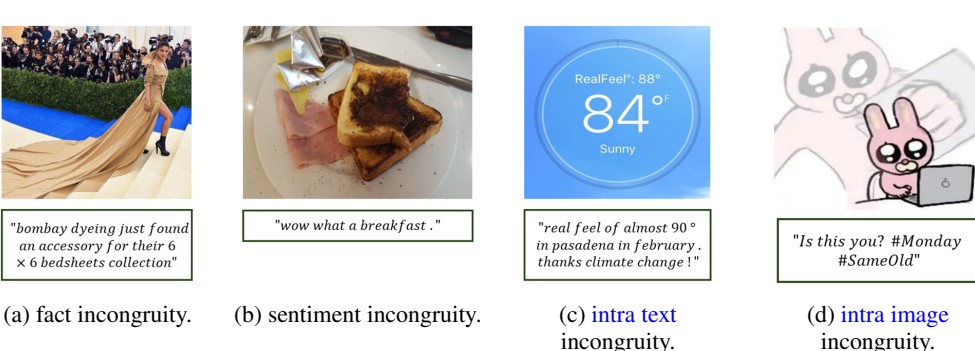

(a) fact incongruity.  (b) sentiment incongruity.  (c) intra text incongruity.  (d) intra image incongruity.

Figure 5: Examples of the incongruity in sarcastic remarks.

**[Sample a in Figure 5] (Fact Incongruity)**
**Reason:** In the text, the image of a woman wearing a floor-length dress is described as "Bombay Dyeing just found an accessory for their 6×6 bedsheets collection." Although the text does not explicitly mention the dress, it metaphorically compares it to a bedsheet. This implicit fact incongruity satirizes the impracticality and excessive extravagance of the woman's attire.

**[Sample b in Figure 5] (Sentiment Incongruity)**
**Reason:** The image shows a burnt piece of bread, which naturally evokes dissatisfaction. In contrast, the text enthusiastically states, "Wow, what a breakfast." This sentiment incongruity highlights the sarcasm of the stark contrast between the quality of the breakfast and the exaggerated expectation.

**[Sample c in Figure 5] (Intra-modal Text Incongruity)**
**Reason:** The image displays a weather forecast with "RealFeel: 88°, Now: 84°, Sunny," which is congruous with the text's statement, "real feel of almost 90° in Pasadena in February." However, the incongruity arises within the text itself: the first sentence conveys dissatisfaction with the unusually high temperature in February, while the second sarcastically states, "thanks climate change." This sentimental reversal highlights the sarcasm of the global warming phenomenon.

**[Sample d in Figure 5] (Intra-modal Image Incongruity)**
**Reason:** The text "#Monday #SameOld" provides contextual background for the image, while "Is this you?" encourages viewers to reflect on the scene. The text complements the image, establishing congruity between the modalities. However, the incongruity within the image arises from the anthropomorphic rabbit appearing to work diligently on a computer while internally imagining smashing it. This contrast satirizes the conflict in modern work environments between outward composure and suppressed frustration.

# B    EXPERIMENTS RESULTS ON MMSD2.0 AND DMSD

To verify the generalization of CMML-Net, we conducted further experiments on the MMSD2.0 and DMSD datasets and report the main results below:

| Modality | Model | ACC(%) | P(%) | R(%) | F1(%) |
|---|---|---|---|---|---|
| Text-Only | TextCNN | 71.61 | 64.62 | 75.22 | 69.52 |
| | BiLSTM | 72.48 | 68.02 | 68.08 | 68.05 |
| | SMSD | 73.56 | 68.45 | 71.55 | 69.97 |
| Image-Only | ResNet | 65.50 | 61.17 | 54.39 | 57.58 |
| | ViT | 72.02 | 65.26 | 74.83 | 69.72 |
| Multimodal | HFM | 70.57 | 64.84 | 69.05 | 66.88 |
| | Att-Bert | 80.03 | 76.28 | 77.82 | 77.04 |
| | CMGCN | 79.83 | 75.82 | 78.01 | 76.90 |
| | HKE | 76.50 | 73.48 | 72.07 | 72.25 |
| | DynRT-Net | 71.40 | 71.80 | 72.17 | 71.34 |
| | Multi-view CLIP (Frozen) | 84.72 | - | - | 83.64 |
| | Multi-view CLIP (Full Finetuned) | 85.64 | 80.33 | 88.24 | 84.10 |
| | **Ours** | **85.83**$^\star$ | **80.58**$^\star$ | **88.30**$^\star$ | **84.26**$^\star$ |

Table 4: Model performance on MMSD2.0 dataset. Results with $\star$ indicate statistical significance over the baseline models at p-value < 0.01. The best results are highlighted in boldface, while the second-best results are underlined.

| Modality | Model | ACC(%) | P(%) | R(%) | F1(%) |
|---|---|---|---|---|---|
| Text-Only | TextCNN | 37.25 | 37.30 | 36.71 | 36.58 |
| | BiLSTM | 34.50 | 33.20 | 32.77 | 32.94 |
| | BERT | 21.25 | 22.22 | 22.28 | 21.25 |
| | RoBERTa | 29.50 | 128.07 | 27.34 | 27.64 |
| Image-Only | ResNet | 28.25 | 27.87 | 27.04 | 27.36 |
| | ViT | 22.00 | 22.53 | 21.36 | 21.55 |
| Multimodal | Res-BERT | 20.75 | 21.62 | 20.77 | 20.60 |
| | Att-Bert | 28.25 | 27.50 | 26.46 | 26.69 |
| | HKE | 37.50 | 37.90 | 37.36 | 37.04 |
| | CMGCN | 34.25 | 35.52 | 35.22 | 34.20 |
| | DMSD-CL | 70.25 | 70.41 | 71.34 | 69.96 |
| | **Ours** | **75.75**$^\star$ | **76.81**$^\star$ | **71.72**$^\star$ | **72.53**$^\star$ |

Table 5: Model performance on DMSD dataset. Results with $\star$ indicate statistical significance over the baseline models at p-value < 0.05. The best results are highlighted in boldface, while the second-best results are underlined.

Due to time constraints, we used the frozen version of CLIP as the encoder. Our CMML-Net still outperforms the fully fine-tuned Multi-view CLIP on MMSD2.0 (Table 4). CMML-Net achieves state-of-the-art (SOTA) results on the DMSD dataset (Table 5).

Based on the results from both the MMSD2.0 and DMSD datasets (Table 4, Table 5), CMML-Net consistently demonstrates its effectiveness in multimodal sarcasm detection. By explicitly and efficiently capturing complete multi-modal incongruities in both fact and sentiment perspectives, our approach achieves competitive performance without requiring extensive fine-tuning or large-scale architectures. This highlights CMML-Net's robustness and strong potential for broader applications in multimodal tasks.

# C  EXPERIMENTAL RESULTS OF MODEL WITH ROBERTA ENCODER ON MSD DATASET

We provide results with RoBERTa as the text encoder, shown in Table 6.

| Model | ACC(%) | Binary-Average | | | Macro-Average | | |
|---|---|---|---|---|---|---|---|
| | | P(%) | R(%) | F1(%) | P(%) | R(%) | F1(%) |
| MILNet | 89.50 | 85.16 | 89.16 | 87.11 | 88.88 | 89.44 | 89.12 |
| DynRT-Net | 93.59 | 93.06 | 93.60 | 93.31 | - | - | - |
| FSICN+RoBERTa | 94.71 | 93.62 | 93.28 | 93.45 | - | - | - |
| **Ours+RoBERTa** | **97.05**⋆ | **99.45**⋆ | 93.29 | **96.27**⋆ | **97.51**⋆ | **96.47**⋆ | **96.92**⋆ |

Table 6: Supplemented results on MSD dataset with RoBERTa encoder. Results with ⋆ indicate statistical significance over the baseline models at p-value < 0.01. The best results are highlighted in boldface, while the second-best results are underlined.

By supplementing these baselines and demonstrating consistently state-of-the-art (SOTA) results (Table 6), our experimental analysis substantiates the superiority of CMML-Net in multimodal sarcasm detection tasks.

# D DISCUSSION OF POTENTIAL APPLICATIONS AND BROADER IMPLICATIONS

Our proposed complete multi-modal metric learning method can jointly and explicitly calculate inter-modal and intra-modal incongruity. It is applicable to multimodal tasks such as fake news detection and sentiment transition analysis It can jointly and efficiently reveal the key incongruous features in these tasks from both inter-modal and intra-modal aspects, thereby improving the performance of the model. This study can provide an important reference for researchers in these similar tasks and promote the further development of multimodal learning methods.

# E  ADDITIONAL RELATED WORK: METRIC LEARNING

Metric learning is a technique for measuring the similarity between objects based on distance metrics. Traditional methods usually transform the original feature space into a representation where the distance can capture meaningful relationships. These methods usually include Mahalanobis distance (Globerson & Roweis, 2005; Wang & Sun, 2015; Weinberger et al., 2005) and rely on linear transformations (e.g., symmetric positive definite matrices) to project the data into Euclidean space. They are limited by their reliance on predefined distance functions.

Deep metric learning extends this concept by leveraging nonlinear transformations through deep neural networks. This approach creates a flexible embedding space. It enables the model to minimize the distance between similar samples and maximize the separation between different samples (Peng et al., 2023; Wang et al., 2023). Deep metric learning has been successful in various applications, including image text retrieval, text classification, face recognition, and multimodal data representation (Suárez et al., 2021). Existing works widely adopt architectures such as Siamese networks and loss functions (e.g., triplet loss (Hoffer & Ailon, 2015) and contrastive loss (Hadsell et al., 2006)) to effectively capture the relationship between pairs or groups of samples.

We introduce the concepts of deep metric learning to calculate incongruity at the representation level rather than at the sample level. Our method iteratively calculates complete multi-modal incongruity to capture the subtle relationships between representations for multi-modal sarcasm detection. This approach generalizes deep metric learning concepts to capture more complex relationships, offering a broader and more adaptable solution for multimodal learning tasks

## F EXPERIMENTAL RESULTS OF MODEL WITH DIFFERENT BACKBONES ON MSD DATASET

To investigate the impact of different backbones of Yolo-task on the MSD, we conducted experiments using various versions of YOLOv10.

Table 7: Results on MSD Dataset with Different Backbones of Yolo-task

| Backbone | ACC (%) | F1 (%) | #Params | FLOPs |
|---|---|---|---|---|
| w/o YOLO-task | 91.19 | 90.85 | - | - |
| YOLO v10-N | 91.61 | 91.27 | 2.3M | 6.7G |
| YOLO v10-M | 91.61 | 91.31 | 15.4M | 59.1G |
| YOLO v10-B | 91.53 | 91.23 | 19.1M | 92.0G |
| YOLO v10-L | 91.53 | 91.24 | 24.4M | 120.3G |
| YOLO v10-X | 91.83 | 91.52 | 29.5M | 160.4G |
| **YOLO v10-S (Ours)** | **92.04** | **91.76** | **7.2M** | **21.6G** |

The experimental results suggest that the performance of the framework is primarily influenced by the guiding role of the YOLO-task, while the complexity of the chosen object detection backbone plays a less significant role.

