# OpenReview forum: "Complete multi-modal metric learning for multi-modal sarcasm detection"
_ICLR.cc/2025/Conference — Submitted to ICLR 2025_

### Official Review · Reviewer_LfU6 · 2024-10-26

**Soundness:** 4
**Presentation:** 3
**Contribution:** 4
**Rating:** 10
**Confidence:** 4

**Summary:**

This paper proposes a complete multi-modal metric learning network (CMML-Net) for multi-modal sarcasm detection tasks. Specifically, CMML-Net utilizes a fact-sentiment multi-task representation learning module to produce refined fact and sentiment text-image representation pairs. It then designs a complete multi-modal metric learning to iteratively calculate inter-modal and intra-modal incongruities in fact and sentiment metric spaces, explicitly capturing complete multi-modal incongruities. CMML-Net performs well in explicitly capturing comprehensive sarcasm information and obtaining discriminative performance via deep metric learning. The state-of-the-art performance on the widely-used dataset demonstrates CMML-Net's effectiveness in multi-modal sarcasm detection.

**Strengths:**

1.The authors use the method of metric learning to study multimodal sarcasm detection, which is innovative.

2.The authors provide a detailed analysis of the effectiveness of each module.

3.The author makes a detailed analysis of the inconsistencies of multimodal irony in images and texts.

**Weaknesses:**

1.Metrics Learning Related Work: This paper is inspired by metrics learning, but lacks work on metrics learning.

2.Typographical errors: There are errors in some of the corner marks in the text, e.g. line 140. Some punctuation errors, such as line 148. Some sentences are redundant, such as lines 165 to 167.

3.Row 153: What is the size of the target range k and whether it will affect the module.

4.Inadequate experimentation: It is not enough to adopt only one dataset, more datasets including MMSD2.0 [1], DMSD [2], RedEval [3] verification model need to be adopted.

5.Supplemental baseline: Comparisons of relevant sarcasm detection work are missing, and it is recommended to add, e.g., G2SAM[4], DynRT-Net[5], DMSD-CL[2].

[1]. MMSD2.0: Towards a Reliable Multi-modal Sarcasm Detection System. ACL-23

[2]. Debiasing Multimodal Sarcasm Detection with Contrastive Learning. AAAI-24

[3]. Leveraging Generative Large Language Models with Visual Instruction and Demonstration Retrieval for Multimodal Sarcasm Detection. NAACL-24.

[4]. G2SAM: Graph-Based Global Semantic Awareness Method for Multimodal Sarcasm Detection. AAAI-24.

[5]. Dynamic Routing Transformer Network for Multimodal Sarcasm Detection. ACL-23.

**Questions:**

None

---

> ### Author Response · Authors · 2024-11-19
>
> **[Q1]** Metrics Learning Related Work: This paper is inspired by metrics learning, but lacks work on metrics learning.
>
> **[A1]** Thank the reviewer for the reviewer’s thorough and insightful comments, which have significantly improved the quality of our work. We will add a summary of the relevant work on "metric learning" in the revised manuscript. The general content of the supplement is shown below.
>
> **Metric learning**
>
> Metric learning is a technique for measuring the similarity between objects based on distance metrics. Traditional methods usually transform the original feature space into a representation where the distance can capture meaningful relationships. These methods usually include Mahalanobis distance [6-8] and rely on linear transformations (e.g., symmetric positive definite matrices) to project the data into Euclidean space. They are limited by their reliance on predefined distance functions.
>
> Deep metric learning extends this concept by leveraging nonlinear transformations through deep neural networks. This approach creates a flexible embedding space. It enables the model to minimize the distance between similar samples and maximize the separation between different samples [2, 3]. Deep metric learning has been successful in various applications, including image text retrieval, text classification, face recognition, and multimodal data representation [1]. Existing works widely adopt architectures such as Siamese networks and loss functions (e.g., triplet loss [4] and contrastive loss [5]) to effectively capture the relationship between pairs or groups of samples.
>
> We introduce the concepts of deep metric learning to calculate incongruity at the representation level rather than at the sample level.  Our method iteratively calculates complete multi-modal incongruity to capture the subtle relationships between representations for multi-modal sarcasm detection. This approach generalizes deep metric learning concepts to capture more complex relationships, offering a broader and more adaptable solution for multimodal learning tasks.
>
> [1] Suárez J L, García S, Herrera F. A tutorial on distance metric learning: Mathematical foundations, algorithms, experimental analysis, prospects and challenges[J]. Neurocomputing, 2021, 425: 300-322.
>
> [2] Peng L, Jian S, Li D, et al. Mrml: Multimodal rumor detection by deep metric learning[C]//ICASSP 2023-2023 IEEE International Conference on Acoustics, Speech and Signal Processing (ICASSP). IEEE, 2023: 1-5.
>
> [3] Wang H, Tang P, Kong H, et al. DHCF: Dual disentangled-view hierarchical contrastive learning for fake news detection on social media[J]. Information Sciences, 2023, 645: 119323.
>
> [4] Hoffer E, Ailon N. Deep metric learning using triplet network[C]//Similarity-based pattern recognition: third international workshop, SIMBAD 2015, Copenhagen, Denmark, October 12-14, 2015. Proceedings 3. Springer International Publishing, 2015: 84-92.
>
> [5] Hadsell R, Chopra S, LeCun Y. Dimensionality reduction by learning an invariant mapping[C]//2006 IEEE computer society conference on computer vision and pattern recognition (CVPR'06). IEEE, 2006, 2: 1735-1742.
>
> [6] Globerson A, Roweis S. Metric learning by collapsing classes[J]. Advances in neural information processing systems, 2005, 18.
>
> [7] Wang F, Sun J. Survey on distance metric learning and dimensionality reduction in data mining[J]. Data mining and knowledge discovery, 2015, 29(2): 534-564.
>
> [8] Weinberger K Q, Saul L K. Distance metric learning for large margin nearest neighbor classification[J]. Journal of machine learning research, 2009, 10(2).
>
> ---
>
> **[Q2]** Typographical errors: There are errors in some of the corner marks in the text, e.g. line 140. Some punctuation errors, such as line 148. Some sentences are redundant, such as lines 165 to 167.
>
> **[A2]** We sincerely thank the reviewer for carefully pointing out typographical errors, punctuation issues, and redundant sentences in the manuscript. These observations are indeed valuable for improving the clarity and presentation of our work.
>
> We will thoroughly review the manuscript to correct the identified issues and perform comprehensive proofreading to ensure overall linguistic accuracy.

---

> ### Author Response · Authors · 2024-11-19
>
> **[Q3]** Row 153: What is the size of the target range k and whether it will affect the module.
>
> **[A3]** We are grateful for the reviewer’s detailed feedback. **k is the dimension of the object existence vector generated by YOLO**. For instance, an object existence vector could be represented as [car: 1, cat: 0, dog: 1, ...]. Since YOLOv10 is pre-trained on the COCO dataset, **the value of k is the same as the number of target categories in the COCO dataset, which is 80**. We can use other object detection models, such as Fast RCNN, to annotate the object existence vector for the training set. But if they are also pre-trained on the COCO dataset, the k value will also be equal to 80.
>
> The main purpose of the YOLO-task is to learn a fine-grained fact representation space that is separated from the sentiment representation space. Therefore, the k value will directly affect the training effect of the YOLO-task on the fact projection layer and may even be misleading. We conducted a simple experiment by randomly selecting k target categories to prove it. The experimental results are shown in the following Table 1.
>
> ### Table 1: Results on MSD Dataset *with different k*
>
> *This table shows the model performances with different k on the MSD dataset.*
>
> | Model | ACC(%) | F1(%) |
> | --- | --- | --- |
> | Ours, k=20 | 91.28 | 90.99 |
> | Ours, k=40 | 91.32  | 91.04  |
> | Ours, k=60 | 91.53 | 91.24 |
> | **Ours, k=80** | **92.04** | **91.76** |
>
> The experimental results confirm the conjecture that the k value will directly affect the training effect of the YOLO-task on the fact projection layer. Therefore, **we recommend choosing an object detection model that is pre-trained on a larger dataset**.
>
> We will make the definition of k more explicit in the revised manuscript.

---

> ### Author Response · Authors · 2024-11-19
>
> **[Q4]** Inadequate experimentation: It is not enough to adopt only one dataset, more datasets including MMSD2.0 [1], DMSD [2], RedEval [3] verification model need to be adopted.
>
> **[A4]** We want to thank the reviewer’s careful review and valuable comments on our work. We conducted further experiments on MMSD2.0 and DMSD and will report the main results of the experiments below (Table 2, Table 3).
>
> ### Table 2: Results on MMSD2.0 Dataset
>
> *This table shows the model performances on the MMSD2.0 dataset.*
>
> | Modality | Model | ACC(%) | P(%) | R(%) | F1(%) |
> | --- | --- | --- | --- | --- | --- |
> | **Text-Only** | TextCNN | 71.61 | 64.62 | 75.22 | 69.52 |
> |  | BiLSTM | 72.48 | 68.02 | 68.08 | 68.05 |
> |  | SMSD | 73.56 | 68.45 | 71.55 | 69.97 |
> | **Image-Only** | Resnet | 65.50 | 61.17 | 54.39 | 57.58 |
> |  | Vit | 72.02 | 65.26 | 74.83 | 69.72 |
> | **Multimodal** | HFM | 70.57 | 64.84 | 69.05 | 66.88 |
> |  | Att-Bert | 80.03 | 76.28 | 77.82 | 77.04 |
> |  | CMGCN | 79.83 | 75.82 | 78.01 | 76.90 |
> |  | HKE | 76.50 | 73.48 | 72.07 | 72.25 |
> |  | DynRT-Net | 71.40 | 71.80 | 72.17 | 71.34 |
> |  | Multi-view CLIP (Frozen) | 84.72 | - | - | 83.64 |
> |  | Multi-view CLIP (Full Finetuned) | 85.64 | 80.33 | 88.24 | 84.10 |
> |  | **Ours** | **85.83*** | **80.58*** | **88.30*** | **84.26*** |
>
> Results with **⋆** denote the significance tests of our CMML-Net over the baseline models except Multi-view CLIP (Full Finetuned) at *p*-value < 0.01. The best results are highlighted in boldface, while the second-best results are underlined.
>
> ### Table 3: Results on DMSD Dataset
>
> *This table shows the model performances on the DMSD dataset.*
>
> | Modality | Model | ACC(%) | P(%) | R(%) | F1(%) |
> | --- | --- | --- | --- | --- | --- |
> | **Text-Only** | TextCNN | 37.25 | 37.30 | 36.71 | 36.58 |
> |  | BiLSTM | 34.50 | 33.20 | 32.77 | 32.94 |
> |  | BERT | 21.25 | 22.22 | 22.28 | 21.25 |
> |  | RoBERTa | 29.50 | 128.07 | 27.34 | 27.64 |
> | **Image-Only** | Resnet | 28.25 | 27.87 | 27.04 | 27.36 |
> |  | Vit | 22.00 | 22.53 | 21.36 | 21.55 |
> | **Multimodal** | Res-BERT | 20.75 | 21.62 | 20.77 | 20.60 |
> |  | Att-Bert | 28.25 | 27.50 | 26.46 | 26.69 |
> |  | HKE | 37.50 | 37.90 | 37.36 | 37.04 |
> |  | CMGCN | 34.25 | 35.52 | 35.22 | 34.20 |
> |  | DMSD-CL | 70.25 | 70.41 | 71.34 | 69.96 |
> |  | **Ours** | **75.75*** | **76.81*** | **71.72*** | **72.53*** |
>
> Results with **⋆** denote the significance tests of our CMML-Net over the baseline models at *p*-value < 0.05. The best results are highlighted in boldface, while the second-best results are underlined.
>
> Due to time constraints, we use the frozen version of CLIP as the encoder. CMML-Net still performs better than the fully fine-tuned Multi-view CLIP on MMSD2.0 (Table 2). Our CMML-Net achieves state-of-the-art (SOTA) results on DMSD (Table 3).
>
> According to the results of the experiment on MMSD2.0 and DMSD (Table 2, Table 3), CMML-Net **consistently demonstrates its effectiveness in multimodal sarcasm detection**. By explicitly and efficiently capturing complete multi-modal incongruities in both fact and sentiment perspectives, our approach achieves competitive performance without requiring extensive fine-tuning or large-scale architectures. This highlights CMML-Net's robustness and strong potential for broader applications in multimodal tasks.
>
> We will add these additional experiments in the revised manuscript to further demonstrate the generalizability of our model.

---

> ### Author Response · Authors · 2024-11-19
>
> **[Q5]** Supplemental baseline: Comparisons of relevant sarcasm detection work are missing, and it is recommended to add, e.g., G2SAM[4], DynRT-Net[5], DMSD-CL[2].
>
> **[A5]** We greatly appreciate the reviewer’s suggestion. We have supplemented the experimental results of G2SAM [1], Multi-view CLIP [2], and DMSD-CL [3] on the MSD dataset. The supplemented content is shown in the following Table 4.
>
> ### Table 4: Supplemented Results on MSD Dataset
>
> *This table shows the model performances on the MSD dataset.*
>
> | Model | ACC (%) | Binary P(%) | Binary R(%) | Binary F1(%) | Macro P(%) | Macro R(%) | Macro F1(%) |
> | --- | --- | --- | --- | --- | --- | --- | --- |
> | G2SAM | 90.48 | 87.95 | 89.02 | 88.48 | 89.44 | 89.79 | 89.65 |
> | Multi-view CLIP (Full Finetuned) | 88.33 | 82.66 | 88.65 | 85.55 | - | - | - |
> | DMSD-CL | 88.95 | 84.89 | 87.90 | 86.37 | 88.35 | 88.77 | 88.54 |
> | **Ours** | **92.04*** | **90.21*** | **90.30*** | **90.25*** | **91.75*** | **91.77*** | **91.76*** |
>
> Results with **⋆** denote the significance tests of our CMML-Net over the baseline models at *p*-value < 0.01. The best results are highlighted in boldface, while the second-best results are underlined.
>
> We list the experimental results of these existing methods which use RoBERTa as the text encoder on the MSD dataset in the following Table 5.
>
> ### Table 5: Supplemented Results on MSD Dataset with RoBERTa encoder
>
> *This table shows the model performances on the MSD dataset.*
>
> | Model | ACC (%) | Binary P(%) | Binary R(%) | Binary F1(%) | Macro P(%) | Macro R(%) | Macro F1(%) |
> | --- | --- | --- | --- | --- | --- | --- | --- |
> | MILNet [4] | 89.50 | 85.16 | 89.16 | 87.11 | 88.88 | 89.44 | 89.12 |
> | DynRT-Net [5] | 93.59 | 93.06 | 93.60 | 93.31 | - | - | - |
> | FSICN+RoBERTa | 94.71 | 93.62 | 93.28 | 93.45 | - | - | - |
> | **Ours+RoBERTa** | **97.05*** | **99.45*** | **93.29** | **96.27*** | **97.51*** | **96.47*** | **96.92*** |
>
> Results with **⋆** denote the significance tests of our CMML-Net over the baseline models at *p*-value < 0.01. The best results are highlighted in boldface, while the second-best results are underlined.
>
> By supplementing these baselines and demonstrating consistently **state-of-the-art (SOTA)** results (Table 4, Table 5), our experimental analysis substantiates the superiority of CMML-Net in multimodal sarcasm detection tasks, both in terms of its methodological innovation and model performance.
>
> We will add these additional experiments in the revised manuscript. Our paper will help readers better understand the existing work in the field of multimodal sarcasm detection.
>
> [1] Wei Y, Yuan S, Zhou H, et al. G^ 2SAM: Graph-Based Global Semantic Awareness Method for Multimodal Sarcasm Detection[C]//Proceedings of the AAAI Conference on Artificial Intelligence. 2024, 38(8): 9151-9159.
>
> [2] Qin L, Huang S, Chen Q, et al. MMSD2. 0: towards a reliable multi-modal sarcasm detection system[J]. arXiv preprint arXiv:2307.07135, 2023.
>
> [3] Jia M, Xie C, Jing L. Debiasing Multimodal Sarcasm Detection with Contrastive Learning[C]//Proceedings of the AAAI Conference on Artificial Intelligence. 2024, 38(16): 18354-18362.
>
> [4] Qiao Y, Jing L, Song X, et al. Mutual-enhanced incongruity learning network for multi-modal sarcasm detection[C]//Proceedings of the AAAI conference on artificial intelligence. 2023, 37(8): 9507-9515.
>
> [5] Tian Y, Xu N, Zhang R, et al. Dynamic routing transformer network for multimodal sarcasm detection[C]//Proceedings of the 61st Annual Meeting of the Association for Computational Linguistics (Volume 1: Long Papers). 2023: 2468-2480.

---

> ### Comment · Reviewer_LfU6 · 2024-11-24
> **reply**
>
> Thanks to the author for the reply, I will revise my score and hope that the author will add the missing details in a future edition.

---

> > ### Author Response · Authors · 2024-11-24
> >
> > We sincerely thank you for your highly professional review work and positive feedback on our work. We will strive to address all the overlooked details in the revised manuscript.

---

### Official Review · Reviewer_TiNN · 2024-10-27

**Soundness:** 2
**Presentation:** 2
**Contribution:** 1
**Rating:** 3
**Confidence:** 5

**Summary:**

This work focus on multimodal sarcasm detection task. As existing works neglected inter-modal or intra-modal incongruities in fact and sentiment perspectives, their performance exist a bias. To achieve this, this paper proposes a complete multi-modal metric learning network (CMML-Net) for multi-modal sarcasm detection tasks. Extensive experiments demonstrates the effectiveness and the scalability of the proposed CMML-Net.

**Strengths:**

1.	The proposed CMML-Net model achieves the state-of-the-art performance on different datsets.
2.	Ablation studies validate the necessity of each component. Visualizations provide intuitive understanding.
3.	The related literatures are well covered.
4.	This work provides code for reproduce.

**Weaknesses:**

1.	Lack of insights in the proposed approach. Motivation of the proposed module in the overall framework is unclear in this paper.
2.	The method of this paper exhibits limited novelty. In my opinion, introducing the Yolo task, the face stream aims to find the image-based incongruity. However, the image-based incongruity have been discussed in existing works. And they proposed many effectiveness approach to solve this problem.[1,2,3,4]
3.	There is unclear motivation of why this paper introduces deep metric learning. And what’s it’s advantage compared to traditional deep learning in existing works?
4.	This paper highlights the incongruity from two perspective: fact and sentiment. The sentiment aspect is easy to understand. However, there lacks more discussion on why the fact aspect is important for multimodal sarcasm detection in introduction section.

[1] H. Liu, W. Wang, and H. Li, “Towards multi-modal sarcasm detection via hierarchical congruity modeling with knowledge enhancement,” in EMNLP, 2022, pp. 4995–5006.
[2] B. Liang, C. Lou, X. Li, L. Gui, M. Yang, and R. Xu, “Multi-modal sarcasm detection with interactive in-modal and cross-modal graphs,” in Proceedings of the ACM International Conference on Multimedia (MM), 2021, pp. 4707–4715.
[3] B. Liang, C. Lou, X. Li, M. Yang, L. Gui, Y. He, W. Pei, and R. Xu, “Multi-modal sarcasm detection via cross-modal graph convolutional network,” in ACL, 2022, pp. 1767–1777.
[4] N. Xu, Z. Zeng, and W. Mao, “Reasoning with multimodal sarcastic tweets via modeling cross-modality contrast and semantic association,” in Proceedings of the Annual Meeting of the Association for Computational Linguistics (ACL), 2020, pp. 3777–3786.

**Questions:**

Please see the weakness.

**Details Of Ethics Concerns:**

There are no ethics concerns.

---

> ### Author Response · Authors · 2024-11-19
>
> **[Q1]** Lack of insights in the proposed approach. Motivation of the proposed module in the overall framework is unclear in this paper.
>
> **[A1]** We would like to reiterate the novelty of our approach. Incongruity is the key clue for multimodal sarcasm detection. However, existing methods **neglected inter-modal or intra-modal incongruities in fact and sentiment perspectives**, leading to incomplete sarcasm information and biased performance. We illustrate this intuitively with several examples and the test results of CMML-Net and different models on several examples in **[CMML-Net Anonymous GitHub](https://anonymous.4open.science/r/CMML-Net-873E/README.md)**.
>
> In multimodal sarcasm detection, fact incongruity reflects contradictions in factual information across modalities, while sentiment incongruity highlights mismatches in emotional tone. We design a multi-task representation learning module to explicitly construct a fact-sentiment dual-stream network. The fact stream (FISN) and the sentiment stream (SISN) **capture fact incongruity and sentiment incongruity respectively** in different representation spaces.
>
> We propose a complete multi-modal metric learning method to **jointly and iteratively calculate inter-modal and intra-modal incongruities** in fact and sentiment metric space. This method avoids the construction of multiple independent architectures for intra-modal and inter-modal incongruities extraction in either the FISN or SISN. Our proposed method is efficient for explicitly capturing complete multi-modal incongruities.
>
> Therefore, our motivation is sufficient and insightful. The test results in **[CMML-Net Anonymous GitHub](https://anonymous.4open.science/r/CMML-Net-873E/README.md)** indicate that our model can **adapt well to inter-modal or intra-modal incongruity in the fact or sentiment perspective**. In addition, our model shows extremely **superior** performance in the main experiment. This also demonstrates the novelty of our method. We present test results here (Table 1) to deal with the connection being inaccessible.
>
> ### Table 1: Test Results on several examples.
>
> *This table shows the test results of CMML-Net and different models on several examples including fact incongruity, sentiment incongruity, intra-modal text incongruity, and intra-modal image incongruity.* **[[CMML-Net Anonymous GitHub](https://anonymous.4open.science/r/CMML-Net-873E/README.md)]**
>
> | **Model** | Fact incongruity Sample (a) | Sentiment incongruity Sample (b) | Intra-modal text incongruity Sample (c) | Intra-modal image incongruity Sample (d) |
> | --- | --- | --- | --- | --- |
> | **HFM** | ✅ | ❌ | ❌ | ❌ |
> | **CMGCN** | ❌ | ✅ | ❌ | ❌ |
> | **Ours** | ✅ | ✅ | ✅ | ✅ |

---

> ### Author Response · Authors · 2024-11-19
>
> **[Q2]** The method of this paper exhibits limited novelty. In my opinion, introducing the Yolo task, the fact stream aims to find the image-based incongruity. However, the image-based incongruity have been discussed in existing works. And they proposed many effectiveness approach to solve this problem.[1,2,3,4]
>
> **[A2]** According to the reviewer’s concerns, we provide a more detailed explanation. Specifically, the fact stream leverages the proposed complete multi-modal metric learning to **jointly and iteratively capture image-based incongruities, text-based incongruities, and inter-modal incongruities**, rather than being limited to detecting only image-based incongruities. We introduce this process in detail in Section 3.2.1.
>
> Then, we emphasize that the main purpose of introducing YOLO-task is **to learn the fact representation space separated from the sentiment stream**. We conduct experiments to replace the backbone of YOLO-task. The experimental results are shown in the following Table 2.
>
> In addition, existing methods use YOLO to detect objects in images and then extract features from the detected objects. They rely heavily on models such as YOLO. However, our YOLO-task only guides the fact projection layer in paying attention to more fine-grained fact semantic information during the training process. It is **removed during reasoning**. Our main result in the paper shows that **our method has a competitive performance advantage**.
>
> ### Table 2: Results on MSD Dataset *with different backbones*
>
> *This table shows the model performances with different backbones on the MSD dataset.*
>
> | Backbone | ACC(%) | F1(%) | #Params  | FLOPs |
> | --- | --- | --- | --- | --- |
> | w/o YOLO-task | 91.19 | 90.85 | - | - |
> | YOLO v10-N | 91.61 | 91.27 | 2.3M | 6.7G |
> | YOLO v10-M | 91.61 | 91.31 | 15.4M | 59.1G |
> | YOLO v10-B | 91.53 | 91.23 | 19.1M | 92.0G |
> | YOLO v10-L | 91.53 | 91.24 | 24.4M | 120.3G |
> | YOLO v10-X | 91.83 | 91.52 | 29.5M | 160.4G |
> | **YOLO v10-S (Ours)** | **92.04** | **91.76** | **7.2M** | **21.6G** |
>
> The experimental results suggest that the performance of the framework is primarily influenced by the guiding role of the YOLO-task, while the complexity of the chosen object detection backbone plays a less significant role.
>
> We will present the purpose of the YOLO task and its difference from other work more clearly in the revised manuscript.
>
> ---
>
> **[Q3]** There is unclear motivation of why this paper introduces deep metric learning. And what’s it’s advantage compared to traditional deep learning in existing works?
>
> **[A3]** We appreciate the reviewer for raising concerns. We propose a deep metric learning method, which is called complete multi-modal metric learning. It can **jointly and iteratively calculate inter-modal and intra-modal incongruity in a unified metric space** (e.g., fact and sentiment metric space). The underlying motivation is to build an efficient model instead of constructing multiple independent architectures in fact and sentiment streams to capture inter-modal and intra-modal incongruity. Deep metric learning has the characteristics of **containing fewer parameters and having a consistent computational structure in the same space**. Therefore, we introduce deep metric learning into multimodal sarcasm detection.
>
> In addition, existing works determine the degree of incongruity between representations before training and then learn the overall effective representation during training. Deep metric learning **measures the incongruity between representations through explicit calculation and learnable distance**. It provides a way to capture complete incongruity more intuitively and effectively than traditional deep learning.
>
> We will strengthen the motivation for introducing deep metric learning and its advantages over traditional deep learning in the revised manuscript.

---

> ### Author Response · Authors · 2024-11-19
>
> **[Q4]** This paper highlights the incongruity from two perspective: fact and sentiment. The sentiment aspect is easy to understand. However, there lacks more discussion on why the fact aspect is important for multimodal sarcasm detection in introduction section.
>
> **[A4]** We apologize for not intuitively clarifying the importance of fact incongruity in the Introduction. According to the reviewer's insightful comments, we provided several specific examples of sarcasm on fact incongruity, sentiment incongruity, and intra-modal incongruity in **[CMML-Net Anonymous GitHub](https://anonymous.4open.science/r/CMML-Net-873E/README.md)**. Additionally, we have included test results of CMML-Net and different models on these examples for further clarity.
>
> The test results clearly show that fact incongruity is useful for sarcasm detection. **Facts describe the existence of objects or events, which are hidden in semantics. Fact incongruity refers to the incongruity between factual semantic information in multi-modal data**. For example, there is a strong factual semantic contrast between a dress and a bedsheet in (a). Our proposed CMML-Net utilizes a fact-sentiment multi-task representation learning module to produce refined fact and sentiment text-image representation pairs. It then explicitly constructs the fact-sentiment dual-stream network to capture incongruity in the fact and sentiment perspectives.
>
> In the revised manuscript, we will include a clearer and more intuitive explanation of the role of fact incongruity in the Introduction. We present test results here again (Table 1) to deal with the connection being inaccessible.
>
> ### Table 1: Test Results on several examples.
>
> *This table shows the test results of CMML-Net and different models on several examples including fact incongruity, sentiment incongruity, intra-modal text incongruity, and intra-modal image incongruity.* **[[CMML-Net Anonymous GitHub](https://anonymous.4open.science/r/CMML-Net-873E/README.md)]**
>
> | **Model** | Fact incongruity Sample (a) | Sentiment incongruity Sample (b) | Intra-modal text incongruity Sample (c) | Intra-modal image incongruity Sample (d) |
> | --- | --- | --- | --- | --- |
> | **HFM** | ✅ | ❌ | ❌ | ❌ |
> | **CMGCN** | ❌ | ✅ | ❌ | ❌ |
> | **Ours** | ✅ | ✅ | ✅ | ✅ |

---

> ### Author Response · Authors · 2024-12-02
> **Reply to reviewer TiNN**
>
> Dear Reviewer TiNN,
>
> We sincerely appreciate your time and effort in reviewing our manuscript and offering valuable suggestions. Since only two days remaining for discussion, we would like to confirm whether our responses have effectively addressed your concern. If you require further clarification, please do not hesitate to contact us. We are more than willing to continue our communication with you. If you are willing to improve the score, we would be grateful.
>
> Thank you for your hard work and support.
>
> Best regards,
>
> The authors of Paper 13962.

---

> ### Comment · Reviewer_TiNN · 2024-12-02
>
> Dear Authors,
>
> Thank you for your detailed response.
>
> I appreciate the effort you put into clarifying the motivation behind your paper. However, I still find it difficult to distinguish your work from prior research in the field. Moreover, considering the feedback from the other reviewers, I will keep my score.
>
> Best regards,
> Reviewer TiNN

---

> > ### Author Response · Authors · 2024-12-03
> >
> > Dear Reviewer:
> >
> > Thanks for your reply. We would like to emphasize our motivations and contributions again as follows.
> >
> > Incongruity is the key clue for multimodal sarcasm detection. However, existing methods **neglected inter-modal or intra-modal incongruities in fact and sentiment perspectives**, leading to incomplete sarcasm information and biased performance. In the link: [[CMML-Net Anonymous GitHub]](https://anonymous.4open.science/r/CMML-Net-873E/README.md) and Table 1, we present several examples to show the importance of these incongruities and show the results of CMML-Net and different models on these examples.
> >
> > In multimodal sarcasm detection, fact incongruity reflects contradictions in factual information across modalities, and sentiment incongruity highlights mismatches in emotional tone. We design a multi-task representation learning module to explicitly construct a fact-sentiment dual-stream network. The fact stream FISN and the sentiment stream SISN respectively **capture fact incongruity and sentiment incongruity in different representation spaces**.
> >
> > In fact and sentiment representation spaces, we design a complete multi-modal metric learning method to **efficiently calculate inter-modal and intra-modal incongruities** within several iterations. This method avoids the construction of multiple independent architectures for extracting intra-modal and inter-modal incongruities in FISN or SISN. Our proposed method performs well in efficiently capturing complete multi-modal incongruities. Our model with 17.3M (about 0.002%) parameters surpasses 7B parameters MLLM on MSD according to the report by Tang et al. (2024) [1]. This result demonstrates our model's effectiveness and efficiency.
> >
> > **Therefore, the most significant difference between our model and previous work is that our model can extract complete incongruity including fact incongruity, sentiment incongruity, inter-modal incongruity, and intra-modal incongruity via an efficient representation-level deep metric learning.  It breaks the performance monopoly of MLLM with about 0.002% parameters and can be well applied in terminals.**
> >
> > Our model achieves competitive performance on MSD, MMSD2.0, and DMSD datasets. According to Tang et al. (2024) [1], our model with 17.3M parameters surpasses LLaVA-1.5-7B (about 500 times) in the supplementary experiments of MMSD2.0. This demonstrates that our work effectively captures complete incongruity and makes an important contribution to multimodal sarcasm detection.
> >
> > Peer reviewer LfU6 gave us great recognition (original score 6 -> latest score 10) after we conducted further detailed experiments and provided sufficient analysis.
> >
> > We sincerely hope that you will also reconsider evaluating our work and improve the score.
> >
> > Best regards,
> > The authors of Paper 13962.
> >
> > ### Table 1: Test Results on several examples.
> >
> > *This table shows the test results of CMML-Net and different models on several examples including fact incongruity, sentiment incongruity, intra-modal text incongruity, and intra-modal image incongruity. HFM [2] utilized inter-modal attention to extra fact incongruity. CMGCN [2] built cross-modal graphs to extract sentiment incongruity.* **[link: [CMML-Net Anonymous GitHub](https://anonymous.4open.science/r/CMML-Net-873E/README.md)]**
> >
> > | **Model** | Fact incongruity Sample (a) **[**[link](https://anonymous.4open.science/r/CMML-Net-873E/files/(a).png)**]** | Sentiment incongruity Sample (b) **[**[link](https://anonymous.4open.science/r/CMML-Net-873E/files/(b).png)**]** | Intra-modal text incongruity Sample (c) **[**[link](https://anonymous.4open.science/r/CMML-Net-873E/files/(c).png)**]** | Intra-modal image incongruity Sample (d) **[**[link](https://anonymous.4open.science/r/CMML-Net-873E/files/(d).png)**]** |
> > | --- | --- | --- | --- | --- |
> > | **HFM** | ✅ | ❌ | ❌ | ❌ |
> > | **CMGCN** | ❌ | ✅ | ❌ | ❌ |
> > | **Ours** | ✅ | ✅ | ✅ | ✅ |
> >
> > [1]. Leveraging Generative Large Language Models with Visual Instruction and Demonstration Retrieval for Multimodal Sarcasm Detection. NAACL-24.
> >
> > [2] Cai Y, Cai H, Wan X. Multi-modal sarcasm detection in twitter with hierarchical fusion model[C]//Proceedings of the 57th annual meeting of the association for computational linguistics. 2019: 2506-2515.
> >
> > [3] Liang B, Lou C, Li X, et al. Multi-modal sarcasm detection via cross-modal graph convolutional network[C]//Proceedings of the 60th Annual Meeting of the Association for Computational Linguistics (Volume 1: Long Papers). Association for Computational Linguistics, 2022, 1: 1767-1777.

---

### Official Review · Reviewer_Trxx · 2024-11-01

**Soundness:** 2
**Presentation:** 2
**Contribution:** 2
**Rating:** 3
**Confidence:** 4

**Summary:**

This paper introduces a novel network, CMML-Net, designed for detecting sarcasm in text-image pairs. The network addresses the limitations of previous methods that focused solely on inter-modal incongruities, neglecting intra-modal incongruities. CMML-Net employs a fact-sentiment multi-task representation learning module to generate refined representations and a complete multi-modal metric learning approach to iteratively calculate inter-modal and intra-modal incongruities in both fact and sentiment metric spaces. The model demonstrates state-of-the-art performance on the Multimodal Sarcasm Detection (MSD) dataset, outperforming existing methods by capturing more comprehensive sarcasm information.

**Strengths:**

1.	This paper is well-written with a clear and concise expression.
2.	The authors have conducted a thorough set of experiments, which is a significant strength of the paper.

**Weaknesses:**

1.	The summary of related work is incomplete.
2.	The authors mention that previous work neglected the importance of intra-modal incongruity in sarcasm detection, leading to incomplete incongruities and biased performance, but do not provide reasons why intra-modal incongruity is useful for sarcasm. It is suggested to use examples in the Introduction section to intuitively demonstrate this.
3.	The authors mention two innovative points in the Introduction section, but these two points essentially seem to be the same.
4.	The authors explain that "Fact incongruity means sarcasm occurs when the literal meaning and the observed facts unexpectedly contrast." Additionally, in the method design, both the FISN and SISN modules take the combined results of image and text as input. This indicates that the work primarily focuses on addressing inter-modal incongruities. However, the authors describe the FISN and SISN as aiming to capture both intra-modal and inter-modal incongruities (Sections 3.2.1 and 3.2.2). Where is the intra-modal incongruity reflected?
5.	Only one dataset is considered in the experiments, such as MMSD2.0 and MSTI, which are not included, and the generalizability of the method cannot be demonstrated.
6.	The Main Result lacks significance analysis.
7.	In Section 4.6, the authors analyze the YOLO-task representation, but is it possible to replace it with other models to achieve multimodal sarcasm detection results based on different backbones?

**Questions:**

Please see the Weaknesses.

---

> ### Author Response · Authors · 2024-11-19
>
> **[Q1]** The summary of related work is incomplete.
>
> **[A1]** Thanks for the reviewer’s thorough and insightful comments, which have significantly improved the quality of our work. We will add a summary of the relevant work on "metric learning" in the revised manuscript. The general content of the supplement is shown below.
>
> **Metric learning**
>
> Metric learning is a technique for measuring the similarity between objects based on distance metrics. Traditional methods usually transform the original feature space into a representation where the distance can capture meaningful relationships. These methods usually include Mahalanobis distance [6-8] and rely on linear transformations (e.g., symmetric positive definite matrices) to project the data into Euclidean space. They are limited by their reliance on predefined distance functions.
>
> Deep metric learning extends this concept by leveraging nonlinear transformations through deep neural networks. This approach creates a flexible embedding space. It enables the model to minimize the distance between similar samples and maximize the separation between different samples [2, 3]. Deep metric learning has been successful in various applications, including image text retrieval, text classification, face recognition, and multimodal data representation [1]. Existing works widely adopt architectures such as Siamese networks and loss functions (e.g., triplet loss [4] and contrastive loss [5]) to effectively capture the relationship between pairs or groups of samples.
>
> We introduce the concepts of deep metric learning to calculate incongruity at the representation level rather than at the sample level.  Our method iteratively calculates complete multi-modal incongruity to capture the subtle relationships between representations for multi-modal sarcasm detection. This approach generalizes deep metric learning concepts to capture more complex relationships, offering a broader and more adaptable solution for multimodal learning tasks.
>
> [1] Suárez J L, García S, Herrera F. A tutorial on distance metric learning: Mathematical foundations, algorithms, experimental analysis, prospects and challenges[J]. Neurocomputing, 2021, 425: 300-322.
>
> [2] Peng L, Jian S, Li D, et al. Mrml: Multimodal rumor detection by deep metric learning[C]//ICASSP 2023-2023 IEEE International Conference on Acoustics, Speech and Signal Processing (ICASSP). IEEE, 2023: 1-5.
>
> [3] Wang H, Tang P, Kong H, et al. DHCF: Dual disentangled-view hierarchical contrastive learning for fake news detection on social media[J]. Information Sciences, 2023, 645: 119323.
>
> [4] Hoffer E, Ailon N. Deep metric learning using triplet network[C]//Similarity-based pattern recognition: third international workshop, SIMBAD 2015, Copenhagen, Denmark, October 12-14, 2015. Proceedings 3. Springer International Publishing, 2015: 84-92.
>
> [5] Hadsell R, Chopra S, LeCun Y. Dimensionality reduction by learning an invariant mapping[C]//2006 IEEE computer society conference on computer vision and pattern recognition (CVPR'06). IEEE, 2006, 2: 1735-1742.
>
> [6] Globerson A, Roweis S. Metric learning by collapsing classes[J]. Advances in neural information processing systems, 2005, 18.
>
> [7] Wang F, Sun J. Survey on distance metric learning and dimensionality reduction in data mining[J]. Data mining and knowledge discovery, 2015, 29(2): 534-564.
>
> [8] Weinberger K Q, Saul L K. Distance metric learning for large margin nearest neighbor classification[J]. Journal of machine learning research, 2009, 10(2).

---

> ### Author Response · Authors · 2024-11-19
>
> **[Q2]** The authors mention that previous work neglected the importance of intra-modal incongruity in sarcasm detection, leading to incomplete incongruities and biased performance, but do not provide reasons why intra-modal incongruity is useful for sarcasm. It is suggested to use examples in the Introduction section to intuitively demonstrate this.
>
> **[A2]** We apologize for not intuitively clarifying the importance of intra-modal incongruity in the Introduction. According to the reviewer's insightful comments, we provided several specific examples of sarcasm on fact incongruity, sentiment incongruity, and intra-modal incongruity in **[CMML-Net Anonymous GitHub](https://anonymous.4open.science/r/CMML-Net-873E/README.md)**. Additionally, we have included test results of CMML-Net and different models on these examples for further clarity.
>
> The test results clearly show that intra-modal incongruity is useful for sarcasm detection. **Incongruity comes from within a modality when the relationship between modalities is congruous**, such as the emphasis relationship in (c) and the complement relationship in (d). Our proposed complete multi-modal metric learning method jointly and iteratively calculates the inter-modal and intra-modal incongruity in fact and sentiment metric space.
>
> In the revised manuscript, we will include a clearer and more intuitive explanation of the role of intra-modal incongruity in the Introduction. We present test results here (Table 1) to deal with the connection being inaccessible.
>
> ### Table 1: Test Results on several examples.
>
> *This table shows the test results of CMML-Net and different models on several examples including fact incongruity, sentiment incongruity, intra-modal text incongruity, and intra-modal image incongruity.* **[[CMML-Net Anonymous GitHub](https://anonymous.4open.science/r/CMML-Net-873E/README.md)]**
>
> | **Model** | Fact incongruity Sample (a) | Sentiment incongruity Sample (b) | Intra-modal text incongruity Sample (c) | Intra-modal image incongruity Sample (d) |
> | --- | --- | --- | --- | --- |
> | **HFM** | ✅ | ❌ | ❌ | ❌ |
> | **CMGCN** | ❌ | ✅ | ❌ | ❌ |
> | **Ours** | ✅ | ✅ | ✅ | ✅ |

---

> ### Author Response · Authors · 2024-11-19
>
> **[Q3]** The authors mention two innovative points in the Introduction section, but these two points essentially seem to be the same.
>
> **[A3]** We apologize for any misunderstanding we may have caused the reviewer. We clarify the differences between the two contributions below.
>
> The first contribution focuses on the proposed deep metric learning method, which is called complete multi-modal metric learning. It can jointly and iteratively **calculate inter-modal and intra-modal incongruity** in fact and sentiment metric space. This deep metric learning method is the key to capturing complete multi-modal incongruity in an efficient model architecture. It avoids the construction of multiple independent architectures in fact or sentiment sub-networks to extract intra-modal and inter-modal incongruity.
>
> The second contribution focuses on the overall structure of CMML-Net. We design multi-task representation learning to explicitly construct a fact-sentiment dual-stream network. The fact-sentiment dual-stream network we constructed effectively **captures incongruity from both fact and sentiment perspectives**, facilitating complete incongruity extraction.
>
> We will further strengthen the presentation of the innovative points in the revised manuscript to make it clearer to readers.
>
> ---
>
> **[Q4]** The authors explain that "Fact incongruity means sarcasm occurs when the literal meaning and the observed facts unexpectedly contrast." Additionally, in the method design, both the FISN and SISN modules take the combined results of image and text as input. This indicates that the work primarily focuses on addressing inter-modal incongruities. However, the authors describe the FISN and SISN as aiming to capture both intra-modal and inter-modal incongruities (Sections 3.2.1 and 3.2.2). Where is the intra-modal incongruity reflected?
>
> **[A4]** We appreciate the reviewer’s insightful question regarding the role of intra-modal incongruity in our design. We clarify that the proposed complete multi-modal metric learning method can jointly and iteratively calculate inter-modal and intra-modal incongruity in FISN and SISN.
>
> Specifically, both FISN and SISN receive text and image representations projected into a unified fact representation space $\mathcal{S}^F$ or sentiment representation space $\mathcal{S}^S$. In this unified space $\mathcal{S}^F$, the complete multi-modal metric learning method iteratively calculates fact incongruity (Eq 3.) between any two representations including intra-modal pairs (e.g., text-text $(\mathbf{r}_T^F, \mathbf{r}_T^F)$ or image-image $(\mathbf{r}_I^F, \mathbf{r}_I^F)$ ) or inter-modal pair (e.g., text-image $(\mathbf{r}_T^F, \mathbf{r}_I^F)$ ):
>
> $\mathcal{I}_{u,v}^F = \| \mathbf{r}_u^F - \mathbf{r}_v^F \|$  (Eq 3)
>
> Here, $u \in set( T, I ) $ and $v \in set( T, I ) $, and $\mathcal{I}_{u,v}^F$ measures the fact incongruity including intra-modal incongruity (e.g., text-text or image-image) and inter-modal incongruity (e.g.,  text-image) in the unified space. **Our method ensures that intra-modal incongruity is fully captured alongside inter-modal incongruity**. We will more clearly demonstrate the extraction of intra-modal incongruity in the revised manuscript.

---

> ### Author Response · Authors · 2024-11-19
>
> **[Q5]** Only one dataset is considered in the experiments, such as MMSD2.0 and MSTI, which are not included, and the generalizability of the method cannot be demonstrated.
>
> **[A5]** Thanks for the reviewer’s careful review and valuable comments. We conducted further experiments on MMSD2.0 and DMSD and will report the main results of the experiments below (Table 2, Table 3).
>
> ### Table 2: Results on MMSD2.0 Dataset
>
> *This table shows the model performances on the MMSD2.0 dataset.*
>
> | Modality | Model | ACC(%) | P(%) | R(%) | F1(%) |
> | --- | --- | --- | --- | --- | --- |
> | **Text-Only** | TextCNN | 71.61 | 64.62 | 75.22 | 69.52 |
> |  | BiLSTM | 72.48 | 68.02 | 68.08 | 68.05 |
> |  | SMSD | 73.56 | 68.45 | 71.55 | 69.97 |
> | **Image-Only** | Resnet | 65.50 | 61.17 | 54.39 | 57.58 |
> |  | Vit | 72.02 | 65.26 | 74.83 | 69.72 |
> | **Multimodal** | HFM | 70.57 | 64.84 | 69.05 | 66.88 |
> |  | Att-Bert | 80.03 | 76.28 | 77.82 | 77.04 |
> |  | CMGCN | 79.83 | 75.82 | 78.01 | 76.90 |
> |  | HKE | 76.50 | 73.48 | 72.07 | 72.25 |
> |  | DynRT-Net | 71.40 | 71.80 | 72.17 | 71.34 |
> |  | Multi-view CLIP (Frozen) | 84.72 | - | - | 83.64 |
> |  | Multi-view CLIP (Full Finetuned) | 85.64 | 80.33 | 88.24 | 84.10 |
> |  | LLaVA 1.5-7B | 85.18 | - | - | - |
> |  | **Ours** | **85.83*** | **80.58*** | **88.30*** | **84.26*** |
>
> Results with **⋆** denote the significance tests of our CMML-Net over the baseline models except Multi-view CLIP (Full Finetuned) at *p*-value < 0.01. The best results are highlighted in boldface, while the second-best results are underlined.
>
> ### Table 3: Results on DMSD Dataset
>
> *This table shows the model performances on the DMSD dataset.*
>
> | Modality | Model | ACC(%) | P(%) | R(%) | F1(%) |
> | --- | --- | --- | --- | --- | --- |
> | **Text-Only** | TextCNN | 37.25 | 37.30 | 36.71 | 36.58 |
> |  | BiLSTM | 34.50 | 33.20 | 32.77 | 32.94 |
> |  | BERT | 21.25 | 22.22 | 22.28 | 21.25 |
> |  | RoBERTa | 29.50 | 128.07 | 27.34 | 27.64 |
> | **Image-Only** | Resnet | 28.25 | 27.87 | 27.04 | 27.36 |
> |  | Vit | 22.00 | 22.53 | 21.36 | 21.55 |
> | **Multimodal** | Res-BERT | 20.75 | 21.62 | 20.77 | 20.60 |
> |  | Att-Bert | 28.25 | 27.50 | 26.46 | 26.69 |
> |  | HKE | 37.50 | 37.90 | 37.36 | 37.04 |
> |  | CMGCN | 34.25 | 35.52 | 35.22 | 34.20 |
> |  | DMSD-CL | 70.25 | 70.41 | 71.34 | 69.96 |
> |  | **Ours** | **75.75*** | **76.81*** | **71.72*** | **72.53*** |
>
> Results with **⋆** denote the significance tests of our CMML-Net over the baseline models at *p*-value < 0.05. The best results are highlighted in boldface, while the second-best results are underlined.
>
> Due to time constraints, we use the frozen version of CLIP as the encoder. CMML-Net still performs better than the fully fine-tuned Multi-view CLIP on MMSD2.0 (Table 2). Our CMML-Net achieves state-of-the-art **(SOTA)** results on DMSD (Table 3).
>
> According to the results of the experiment on MMSD2.0 and DMSD (Table 2, Table 3), CMML-Net **consistently demonstrates its effectiveness** in multimodal sarcasm detection. By explicitly and efficiently capturing complete multi-modal incongruities in both fact and sentiment perspectives, our approach achieves competitive performance without requiring extensive fine-tuning or large-scale architectures. This highlights CMML-Net's robustness and strong potential for broader applications in multimodal tasks.
>
> We will add these additional experiments in the revised manuscript to further demonstrate the generalizability of our model.

---

> ### Author Response · Authors · 2024-11-19
>
> **[Q6]** The Main Result lacks significance analysis.
>
> **[A6]** We thank the reviewer for identifying this oversight. We randomly sampled several random number seeds to run the model and conduct significance tests. In the Main Result of **MSD**, the significance test indicates that **all metrics of CMML-Net outperform all baseline models selected at *p*-value < 0.01**. We will add this additional experiment in the revised manuscript to further demonstrate the robustness of our model.
>
> ---
>
> **[Q7]** In Section 4.6, the authors analyze the YOLO-task representation, but is it possible to replace it with other models to achieve multimodal sarcasm detection results based on different backbones?
>
> **[A7]** We are grateful for the reviewer’s detailed feedback. The main purpose of the YOLO-task is to train the fact projection layer to learn a fine-grained fact representation space separated from the emotion representation space. **YOLO does not participate in reasoning**. Therefore, we can use other object detection models, such as Fast RCNN, to annotate the object existence vector for the training set. For instance, an object existence vector could be represented as [car: 1, cat: 0, dog: 1, ...].
>
> Due to time constraints, we conducted simple experiments using different parameter versions of YOLOv10. The experimental results are shown in the following Table 4. They reveal that the performance improvement is more due to the separation of the representation space and the fact-sentiment dual-stream network.
>
> ### Table 4: Results on MSD Dataset *with different backbones*
>
> *This table shows the model performances with different backbones on the MSD dataset.*
>
> | Backbone | ACC(%) | F1(%) | #Params | FLOPs |
> | --- | --- | --- | --- | --- |
> | w/o YOLO-task | 91.19 | 90.85 | - | - |
> | YOLO v10-N | 91.61 | 91.27 | 2.3M | 6.7G |
> | YOLO v10-M | 91.61 | 91.31 | 15.4M | 59.1G |
> | YOLO v10-B | 91.53 | 91.23 | 19.1M | 92.0G |
> | YOLO v10-L | 91.53 | 91.24 | 24.4M | 120.3G |
> | YOLO v10-X | 91.83 | 91.52 | 29.5M | 160.4G |
> | **YOLO v10-S (Ours)** | **92.04** | **91.76** | **7.2M** | **21.6G** |
>
> The experimental results suggest that the performance of the framework is primarily influenced by the guiding role of the YOLO-task, **while the complexity of the chosen object detection backbone plays a less significant role**. We will add this additional experiment in the revised manuscript to further show the role of YOLO-task.

---

> ### Author Response · Authors · 2024-12-02
> **Reply to reviewer Trxx**
>
> Dear Reviewer Trxx,
>
> We sincerely appreciate your time and effort in reviewing our manuscript and offering valuable suggestions. Since only two days remaining for discussion, we would like to confirm whether our responses have effectively addressed your concern. If you require further clarification, please do not hesitate to contact us. We are more than willing to continue our communication with you. If you are willing to improve the score, we would be grateful.
>
> Thank you for your hard work and support.
>
> Best regards,
>
> The authors of Paper 13962.

---

> > ### Comment · Reviewer_Trxx · 2024-12-02
> >
> > Dear Authors,
> >
> > Thank you very much for your response.\
> > Although you have addressed some of my concerns in the response, considering the similarity between this paper and previous work, as well as the research motivation of this work. I still tend to reject this paper. It is suggested that the author clarify the differences from previous work and highlight the innovation of this work in subsequent revisions, otherwise, this work may appear very incremental.
> >
> > Best,\
> > Reviewer Trxx

---

> > > ### Author Response · Authors · 2024-12-03
> > >
> > > Dear Reviewer:
> > >
> > > Thanks for your reply. We would like to emphasize our motivations and contributions again as follows.
> > >
> > > Incongruity is the key clue for multimodal sarcasm detection. However, existing methods **neglected inter-modal or intra-modal incongruities in fact and sentiment perspectives**, leading to incomplete sarcasm information and biased performance. In the link: [[CMML-Net Anonymous GitHub]](https://anonymous.4open.science/r/CMML-Net-873E/README.md) and Table 1, we present several examples to show the importance of these incongruities and show the results of CMML-Net and different models on these examples.
> > >
> > > In multimodal sarcasm detection, fact incongruity reflects contradictions in factual information across modalities, and sentiment incongruity highlights mismatches in emotional tone. We design a multi-task representation learning module to explicitly construct a fact-sentiment dual-stream network. The fact stream FISN and the sentiment stream SISN respectively **capture fact incongruity and sentiment incongruity in different representation spaces**.
> > >
> > > In fact and sentiment representation spaces, we design a complete multi-modal metric learning method to **efficiently calculate inter-modal and intra-modal incongruities** within several iterations. This method avoids the construction of multiple independent architectures for extracting intra-modal and inter-modal incongruities in FISN or SISN. Our proposed method performs well in efficiently capturing complete multi-modal incongruities. Our model with 17.3M (about 0.002%) parameters surpasses 7B parameters MLLM on MSD according to the report by Tang et al. (2024) [1]. This result demonstrates our model's effectiveness and efficiency.
> > >
> > > **Therefore, the most significant difference between our model and previous work is that our model can extract complete incongruity including fact incongruity, sentiment incongruity, inter-modal incongruity, and intra-modal incongruity via an efficient representation-level deep metric learning.  It breaks the performance monopoly of MLLM with about 0.002% parameters and can be well applied in terminals.**
> > >
> > > Our model achieves competitive performance on MSD, MMSD2.0, and DMSD datasets. According to Tang et al. (2024) [1], our model with 17.3M parameters surpasses LLaVA-1.5-7B (about 500 times) in the supplementary experiments of MMSD2.0. This demonstrates that our work effectively captures complete incongruity and makes an important contribution to multimodal sarcasm detection.
> > >
> > > Peer reviewer LfU6 gave us great recognition (original score 6 -> latest score 10) after we conducted further detailed experiments and provided sufficient analysis.
> > >
> > > We sincerely hope that you will also reconsider evaluating our work and improve the score.
> > >
> > > Best regards,
> > > The authors of Paper 13962.
> > >
> > > ### Table 1: Test Results on several examples.
> > >
> > > *This table shows the test results of CMML-Net and different models on several examples including fact incongruity, sentiment incongruity, intra-modal text incongruity, and intra-modal image incongruity. HFM [2] utilized inter-modal attention to extra fact incongruity. CMGCN [2] built cross-modal graphs to extract sentiment incongruity.* **[link: [CMML-Net Anonymous GitHub](https://anonymous.4open.science/r/CMML-Net-873E/README.md)]**
> > >
> > > | **Model** | Fact incongruity Sample (a) **[**[link](https://anonymous.4open.science/r/CMML-Net-873E/files/(a).png)**]** | Sentiment incongruity Sample (b) **[**[link](https://anonymous.4open.science/r/CMML-Net-873E/files/(b).png)**]** | Intra-modal text incongruity Sample (c) **[**[link](https://anonymous.4open.science/r/CMML-Net-873E/files/(c).png)**]** | Intra-modal image incongruity Sample (d) **[**[link](https://anonymous.4open.science/r/CMML-Net-873E/files/(d).png)**]** |
> > > | --- | --- | --- | --- | --- |
> > > | **HFM** | ✅ | ❌ | ❌ | ❌ |
> > > | **CMGCN** | ❌ | ✅ | ❌ | ❌ |
> > > | **Ours** | ✅ | ✅ | ✅ | ✅ |
> > >
> > > [1]. Leveraging Generative Large Language Models with Visual Instruction and Demonstration Retrieval for Multimodal Sarcasm Detection. NAACL-24.
> > >
> > > [2] Cai Y, Cai H, Wan X. Multi-modal sarcasm detection in twitter with hierarchical fusion model[C]//Proceedings of the 57th annual meeting of the association for computational linguistics. 2019: 2506-2515.
> > >
> > > [3] Liang B, Lou C, Li X, et al. Multi-modal sarcasm detection via cross-modal graph convolutional network[C]//Proceedings of the 60th Annual Meeting of the Association for Computational Linguistics (Volume 1: Long Papers). Association for Computational Linguistics, 2022, 1: 1767-1777.

---

### Official Review · Reviewer_BJ3o · 2024-11-04

**Soundness:** 2
**Presentation:** 2
**Contribution:** 2
**Rating:** 5
**Confidence:** 3

**Summary:**

The paper introduces a novel framework called the Complete Multi-Modal Metric Learning Network (CMML-Net) designed for multi-modal sarcasm detection, leveraging both text and image data. This task is complex due to sarcasm's reliance on implicit contrasts, often between literal meanings and actual sentiments or facts. The CMML-Net improves sarcasm detection by identifying these contrasts through inter-modal (between text and image) and intra-modal (within each modality) incongruities, enhancing sarcasm recognition accuracy. In general, the CMML-Net demonstrates a significant advancement in multi-modal sarcasm detection, providing a repeatable and well-organized structure for detecting sarcasm with high accuracy. The model’s design effectively captures multi-dimensional incongruities, though its computational demands and current scope might limit broader real-time and cross-modal applications.

**Strengths:**

1. The modular structure of CMML-Net enables clear, systematic analysis of sarcasm, making the model robust and extensible for future research. The dual-stream network is meticulously designed to assess sarcasm through both fact and sentiment incongruities, improving detection accuracy.

2. By building upon existing work and leveraging well-established models, the CMML-Net is highly repeatable, with well-documented performance on benchmark datasets.

**Weaknesses:**

1. The paper lacks clarity due to some undefined symbols and terms. For instance, the architecture is based on "units" in section 3.2 that are repeatedly referenced as learnable, but their exact nature is not defined. It is unclear whether these units are neuron clusters, specific network layers, or memory mechanisms designed to integrate multiple modalities. Additionally, other symbols, such as the capital "S" and "F," are not explicitly defined. While "S" seems to represent sentiment-related information, the meaning of "F" is ambiguous and should be clarified. Could you please clarify those terms where they are firstly introduced?

2.  The paper’s focus on multi-modal sarcasm detection is narrow, limiting its potential impact and relevance for the broader machine learning community. Could you please give some potential broader implications or applications of their work beyond sarcasm detection.
3.  The framework is built largely on existing approaches, enhancing its reproducibility. However, this reliance on established methodologies limits its originality, as there is a lack of significant methodological contribution. This may affect its impact within the research community, which typically values innovation in addition to reproducibility.

**Questions:**

See in weaknesses.

---

> ### Author Response · Authors · 2024-11-19
>
> **[Q1]** The paper lacks clarity due to some undefined symbols and terms. For instance, the architecture is based on "units" in section 3.2 that are repeatedly referenced as learnable, but their exact nature is not defined. It is unclear whether these units are neuron clusters, specific network layers, or memory mechanisms designed to integrate multiple modalities. Additionally, other symbols, such as the capital "S" and "F," are not explicitly defined. While "S" seems to represent sentiment-related information, the meaning of "F" is ambiguous and should be clarified. Could you please clarify those terms where they are firstly introduced?
>
> **[A1]** We acknowledge that some terms and symbols are not fully defined, and we apologize for any confusion caused. We will address the specific issues in the revised paper:
>
> The CMML Unit in Section 3.2 is the designed **basic component** for complete multi-modal metric learning in the fact incongruity sub-network(FISN) and sentiment incongruity sub-network(SISN). It calculates the incongruity of all text-text, text-image, and image-image representation pairs in a unified metric space, and updates them via dynamic separation and non-linear adjustment to obtain more discriminative incongruity representations. FISN and SISN will gradually obtain complete multi-modal incongruities by stacking CMML Units.
>
> The superscript capital letters "F" and "S" stand for **Fact** and **Sentiment**, respectively. They distinguish the different learning processes for fact and sentiment representations in the multi-task learning module and the calculations in the fact and sentiment streams of the dual-stream network.
>
> We thank the reviewer for the careful review work, and we will revise the manuscript to ensure that all terms and symbols are clearly defined and unambiguous.
>
> ---
>
> **[Q2]** The paper’s focus on multi-modal sarcasm detection is narrow, limiting its potential impact and relevance for the broader machine learning community. Could you please give some potential broader implications or applications of their work beyond sarcasm detection.
>
> **[A2]** Thanks for the reviewer’s thorough and insightful comments. This study focuses on multimodal sarcasm detection, but we believe that the proposed model and method have broader application potential.
>
> Our proposed complete multi-modal metric learning method can jointly and explicitly calculate inter-modal and intra-modal incongruity. It is applicable to multimodal tasks such as **fake news detection and sentiment transition analysis** It can **jointly and efficiently reveal the key incongruous features in these tasks from both inter-modal and intra-modal aspects**, thereby improving the performance of the model. This study can provide an important reference for researchers in these similar tasks and promote the further development of multimodal learning methods.
>
> We will add a discussion of potential broader implications and applications of CMML-Net in the revised manuscript.

---

> ### Author Response · Authors · 2024-11-19
>
> **[Q3]** The framework is built largely on existing approaches, enhancing its reproducibility. However, this reliance on established methodologies limits its originality, as there is a lack of significant methodological contribution. This may affect its impact within the research community, which typically values innovation in addition to reproducibility.
>
> **[A3]** We acknowledge that we stand on the shoulders of giants, but we believe that this work is novel and contributes to the field of multimodal satirical detection.
>
> Incongruity is the key clue for multimodal sarcasm detection. However, existing methods **neglected inter-modal or intra-modal incongruities in fact and sentiment perspectives**, leading to incomplete sarcasm information and biased performance. In **[CMML-Net Anonymous GitHub](https://anonymous.4open.science/r/CMML-Net-873E/README.md)**, we present several examples to show the importance of these incongruities and show the results of CMML-Net and different models on these examples.
>
> In multimodal sarcasm detection, fact incongruity reflects contradictions in factual information across modalities, and sentiment incongruity highlights mismatches in emotional tone. We design a multi-task representation learning module to explicitly construct a fact-sentiment dual-stream network. The fact stream FISN and the sentiment stream SISN respectively **capture fact incongruity and sentiment incongruity in different representation spaces**.
>
> Moreover, we propose a complete multi-modal metric learning method to **jointly and iteratively calculate inter-modal and intramodal incongruities in a unified metric space** (e.g., fact and sentiment metric space). This method avoids the construction of multiple independent architectures for extracting intra-modal and inter-modal incongruities in FISN or SISN. Our proposed method performs well in explicitly capturing complete multi-modal incongruities.
>
> In experiments, the test results in **[CMML-Net Anonymous GitHub](https://anonymous.4open.science/r/CMML-Net-873E/README.md)** indicate that our model can adapt well to inter-modal or intra-modal incongruity in the fact or sentiment perspective. Furthermore, our model shows extremely superior performance in the main results. This also demonstrates the novelty of our method. Therefore, our work is novel and contributes to multimodal sarcasm detection. We present test results here (Table 1) to deal with the connection being inaccessible.
>
> ### Table 1: Test Results on several examples.
>
> *This table shows the test results of CMML-Net and different models on several examples including fact incongruity, sentiment incongruity, intra-modal text incongruity, and intra-modal image incongruity.* **[[CMML-Net Anonymous GitHub](https://anonymous.4open.science/r/CMML-Net-873E/README.md)]**
>
> | **Model** | Fact incongruity Sample (a) | Sentiment incongruity Sample (b) | Intra-modal text incongruity Sample (c) | Intra-modal image incongruity Sample (d) |
> | --- | --- | --- | --- | --- |
> | **HFM** | ✅ | ❌ | ❌ | ❌ |
> | **CMGCN** | ❌ | ✅ | ❌ | ❌ |
> | **Ours** | ✅ | ✅ | ✅ | ✅ |

---

> ### Author Response · Authors · 2024-12-02
> **Reply to reviewer BJ3o**
>
> Dear Reviewer BJ3o,
>
> We sincerely appreciate your time and effort in reviewing our manuscript and offering valuable suggestions. Since only two days remaining for discussion, we would like to confirm whether our responses have effectively addressed your concern. If you require further clarification, please do not hesitate to contact us. We are more than willing to continue our communication with you. If you are willing to improve the score, we would be grateful.
>
> Thank you for your hard work and support.
>
> Best regards,
>
> The authors of Paper 13962.

---

> > ### Comment · Reviewer_BJ3o · 2024-12-02
> > **RE**
> >
> > Thank you for the information. However, as several other reviewers have noted and the comments from my previous response, the main idea of this paper remains within the bounds of the existing framework. Therefore, I will maintain my current score.

---

> > > ### Author Response · Authors · 2024-12-03
> > >
> > > Dear Reviewer:
> > >
> > > Thanks for your reply. We would like to emphasize our motivations and contributions again as follows.
> > >
> > > Incongruity is the key clue for multimodal sarcasm detection. However, existing methods **neglected inter-modal or intra-modal incongruities in fact and sentiment perspectives**, leading to incomplete sarcasm information and biased performance. In the link: [[CMML-Net Anonymous GitHub]](https://anonymous.4open.science/r/CMML-Net-873E/README.md) and Table 1, we present several examples to show the importance of these incongruities and show the results of CMML-Net and different models on these examples.
> > >
> > > In multimodal sarcasm detection, fact incongruity reflects contradictions in factual information across modalities, and sentiment incongruity highlights mismatches in emotional tone. We design a multi-task representation learning module to explicitly construct a fact-sentiment dual-stream network. The fact stream FISN and the sentiment stream SISN respectively **capture fact incongruity and sentiment incongruity in different representation spaces**.
> > >
> > > In fact and sentiment representation spaces, we design a complete multi-modal metric learning method to **efficiently calculate inter-modal and intra-modal incongruities** within several iterations. This method avoids the construction of multiple independent architectures for extracting intra-modal and inter-modal incongruities in FISN or SISN. Our proposed method performs well in efficiently capturing complete multi-modal incongruities. Our model with 17.3M (about 0.002%) parameters surpasses 7B parameters MLLM on MSD according to the report by Tang et al. (2024) [1]. This result demonstrates our model's effectiveness and efficiency.
> > >
> > > **Therefore, the most significant difference between our model and previous work is that our model can extract complete incongruity including fact incongruity, sentiment incongruity, inter-modal incongruity, and intra-modal incongruity via an efficient representation-level deep metric learning.  It breaks the performance monopoly of MLLM with about 0.002% parameters and can be well applied in terminals.**
> > >
> > > Our model achieves competitive performance on MSD, MMSD2.0, and DMSD datasets. According to Tang et al. (2024) [1], our model with 17.3M parameters surpasses LLaVA-1.5-7B (about 500 times) in the supplementary experiments of MMSD2.0. This demonstrates that our work effectively captures complete incongruity and makes an important contribution to multimodal sarcasm detection.
> > >
> > > Peer reviewer LfU6 gave us great recognition (original score 6 -> latest score 10) after we conducted further detailed experiments and provided sufficient analysis.
> > >
> > > We sincerely hope that you will also reconsider evaluating our work and improve the score.
> > >
> > > Best regards,
> > > The authors of Paper 13962.
> > >
> > > ### Table 1: Test Results on several examples.
> > >
> > > *This table shows the test results of CMML-Net and different models on several examples including fact incongruity, sentiment incongruity, intra-modal text incongruity, and intra-modal image incongruity. HFM [2] utilized inter-modal attention to extra fact incongruity. CMGCN [2] built cross-modal graphs to extract sentiment incongruity.* **[link: [CMML-Net Anonymous GitHub](https://anonymous.4open.science/r/CMML-Net-873E/README.md)]**
> > >
> > > | **Model** | Fact incongruity Sample (a) **[**[link](https://anonymous.4open.science/r/CMML-Net-873E/files/(a).png)**]** | Sentiment incongruity Sample (b) **[**[link](https://anonymous.4open.science/r/CMML-Net-873E/files/(b).png)**]** | Intra-modal text incongruity Sample (c) **[**[link](https://anonymous.4open.science/r/CMML-Net-873E/files/(c).png)**]** | Intra-modal image incongruity Sample (d) **[**[link](https://anonymous.4open.science/r/CMML-Net-873E/files/(d).png)**]** |
> > > | --- | --- | --- | --- | --- |
> > > | **HFM** | ✅ | ❌ | ❌ | ❌ |
> > > | **CMGCN** | ❌ | ✅ | ❌ | ❌ |
> > > | **Ours** | ✅ | ✅ | ✅ | ✅ |
> > >
> > > [1]. Leveraging Generative Large Language Models with Visual Instruction and Demonstration Retrieval for Multimodal Sarcasm Detection. NAACL-24.
> > >
> > > [2] Cai Y, Cai H, Wan X. Multi-modal sarcasm detection in twitter with hierarchical fusion model[C]//Proceedings of the 57th annual meeting of the association for computational linguistics. 2019: 2506-2515.
> > >
> > > [3] Liang B, Lou C, Li X, et al. Multi-modal sarcasm detection via cross-modal graph convolutional network[C]//Proceedings of the 60th Annual Meeting of the Association for Computational Linguistics (Volume 1: Long Papers). Association for Computational Linguistics, 2022, 1: 1767-1777.

---

### Meta-Review · Area_Chair_SQiq · 2024-12-20

**Metareview:**

This paper proposes a metric learning framework to enhance the task of multimodal sarcasm detection. By combining two aspects, namely sentiment aspect and factual aspect, the proposed method incorporates intra- and inter-modal incongruities across two aspects to capture more nuanced samples. Experiments demonstrate the proposed method outperforms existing baselines on the MSD dataset.

Strengths:
- The joint model which aims to capture both sentiment-oriented and fact-oriented sarcasm is reasonable and interesting. Despite the fact that inter-modal and intra-modal relationships have been proposed earlier, the integration of two perspectives is novel.
- The adoption of metric learning for this task is efficient and effective.
- Experiments on the MSD dataset shows the proposed method gives the best performance.

Weaknesses:
- The contribution is limited. Several existing methods have incorporated both inter-modal and intra-modal relationships for sarcasm detection (e.g., Pan et al. 2020, Liang et al. 2021). It is not accurate to claim this contribution.
- The initial experiment only considers one dataset with slight improvement. Although the authors provided more experiments with other datasets as suggested by reviewers, the result does not seem to indicate substantial gains, especially for MMSD 2.0.
- The motivation of the model design is lacking. It is not very clear why metric learning is beneficial in this task, compared with existing deep learning methods, such as using a neural incongruity scorer.
- Most importantly, the application domain is narrow. More diverse tasks such as fake news detection should be included to demonstrate the impact of the proposed method.

**Additional Comments On Reviewer Discussion:**

- Reviewers have raised questions regarding significance test, other benchmark datasets and baseline models. In response, the authors provided additional experiments to address these concerns.
- Reviewers also raised questions about changing different backbone models. The authors performed additional experiments using different versions of YOLO, but not other completely different backbones, which is not sufficient to address this concern.
- Despite the effort, some critical concerns remain unsolved, including: (1) the limited novelty of the methodology; (2) the limited scope of the method to other application domains; (3) the lack of clearer motivation of using metric learning.

---

### Decision · Program_Chairs · 2025-01-22

Reject